# Contrastive Lift: 3D Object Instance Segmentation by Slow-Fast Contrastive Fusion

**Yash Bhalgat**   **Iro Laina**   **João F. Henriques**   **Andrew Zisserman**   **Andrea Vedaldi**

Visual Geometry Group
University of Oxford
{yashsb,iro,joao,az,vedaldi}@robots.ox.ac.uk

## Abstract

Instance segmentation in 3D is a challenging task due to the lack of large-scale annotated datasets. In this paper, we show that this task can be addressed effectively by leveraging instead 2D pre-trained models for instance segmentation. We propose a novel approach to lift 2D segments to 3D and fuse them by means of a neural field representation, which encourages multi-view consistency across frames. The core of our approach is a *slow-fast* clustering objective function, which is scalable and well-suited for scenes with a large number of objects. Unlike previous approaches, our method does not require an upper bound on the number of objects or object tracking across frames. To demonstrate the scalability of the slow-fast clustering, we create a new semi-realistic dataset called the Messy Rooms dataset, which features scenes with up to 500 objects per scene. Our approach outperforms the state-of-the-art on challenging scenes from the ScanNet, Hypersim, and Replica datasets, as well as on our newly created Messy Rooms dataset, demonstrating the effectiveness and scalability of our slow-fast clustering method.

## 1   Introduction

While the content of images is three-dimensional, image understanding has largely developed by treating images as two-dimensional patterns. This was primarily due to the lack of effective machine learning tools that could model content in 3D. However, recent advancements in neural field methods [5, 40, 42, 49, 67] have provided an effective approach for applying deep learning to 3D signals. These breakthroughs enable us to revisit image understanding tasks in 3D, accounting for factors such as multi-view consistency and occlusions.

In this paper, we study the problem of *object instance segmentation* in 3D. Our goal is to extend 2D instance segmentation to the third dimension, enabling simultaneous 3D reconstruction and 3D instance segmentation. Our approach is to extract information from multiple views of a scene independently with a pre-trained 2D instance segmentation model and fuse it into a single 3D neural field. Our main motivation is that, while acquiring densely labelled 3D datasets is challenging, annotations and pre-trained predictors for 2D data are widely available. Recent approaches have also capitalized on this idea, demonstrating their potential for 2D-to-3D semantic segmentation [31, 37, 60, 67] and distilling general-purpose 2D features in 3D space [29, 55]. When distilling semantic labels or features, the information to be fused is inherently consistent across multiple views: semantic labels are viewpoint invariant, and 2D features across views are typically learned with the same loss function. Additionally, the number of labels or feature dimensions is predetermined. Thus, 3D fusion amounts to multi-view aggregation.

When it comes to instance segmentation, however, the number of objects in a 3D scene is not fixed or known, and can indeed be quite large compared to the number of semantic classes. More importantly, when objects are detected independently in different views, they are assigned different

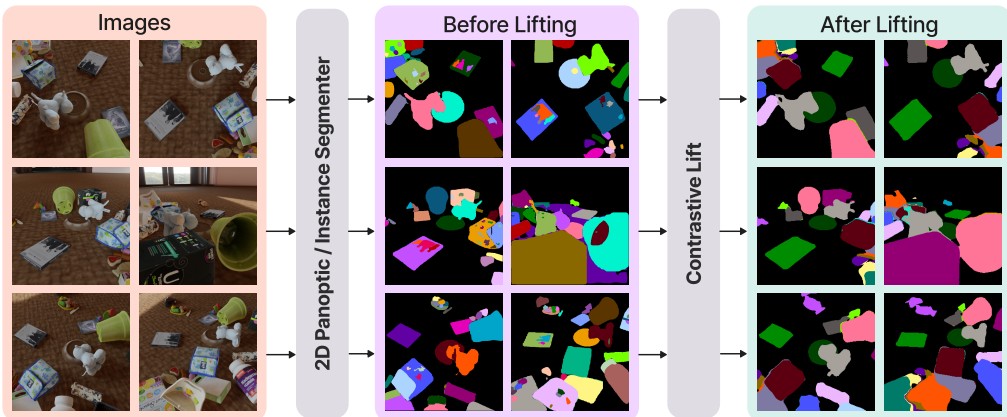

Figure 1: **Contrastive Lift** takes as input several views of a scene (left), as well as the output of a panoptic 2D segmenter (middle). It then reconstructs the scene in 3D while fusing the 2D segments, which are noisy and generally labelled inconsistently between views, when no object association (tracking) is assumed. Our method represents object instances in 3D space by a low-dimensional continuous embedding which can be trained efficiently using a contrastive formulation that is agnostic to the inconsistent labelling across views. The result (right) is a consistent 3D segmentation of the objects, which, once imaged, results in more accurate and consistent 2D segmentations.

and inconsistent identifiers, which cannot be aggregated directly. The challenge is thus how to fuse information that is not presented in a viewpoint-consistent manner.

Recently, Panoptic Lifting [48] proposed to resolve the lack of multi-view consistency by explicitly fitting a permutation that aligns labels extracted from multiple views. Although this yields good results, there are two drawbacks to this approach. Firstly, determining the permutation matrix involves solving a linear assignment problem using Hungarian Matching for every gradient computation. The cost of this increases cubically with the number of identifiers, which may limit scalability when dealing with a large number of object instances. Secondly, the canonical label space, where the permutation maps each 2D label, may need to be extensive to accommodate a large number of objects.

In this study, we propose a more efficient formulation, which also leads to more accurate results. To understand our approach, consider first a 2D image segmenter: it takes an image $I$ as input and produces a mapping $y$ that assigns each pixel $u \in \mathbb{R}^2$ to an object instance label $y(u) \in \{1, \ldots, L\}$. It is natural to extend this mapping to 3D by introducing a function $Y$ that associates each 3D point $x \in \mathbb{R}^3$ with the label $Y(x)$ of the corresponding object. To account for the fact that labels are arbitrary and thus inconsistent between views, Panoptic Lifting [48] seeks an image-dependent permutation matrix $P$ such that $Y(x) = P \cdot y(u)$, where $u$ is the projection of $x$ onto the image.

To address the aforementioned challenges with the linear-assignment-based approach, we identify the labels $y(u)$ with coordinate vectors in the Euclidean space $\mathbb{R}^L$. The functions $y(u)$ can be reconstructed, up to a label permutation, from the distances $d(y(u), y(u')) = \|y(u) - y(u')\|_2$ of such vectors, as they tell whether labels of two pixels $(u, u')$ are the same or different, without considering the specific labelling. Notably, similar to compressed sensing, we can seek lower-dimensional projections of the vectors $y$ that preserve this information. With this in mind, we replace the 3D labelling function $Y$ with a low-dimensional Euclidean embedding $\Theta(x) \in \mathbb{R}^D$. Then, we supervise the embeddings such that their distances $d(\Theta(x), \Theta(x')) \approx d(y(u), y(u'))$ are sufficiently similar to that of corresponding 2D label embeddings.

This approach has two advantages. First, it only requires learning vectors of dimensionality $D \ll L$ which is independent of the number of objects $L$. Second, learning this function does not require solving an assignment problem; rather, it only considers pairwise distances. Hence, the complexity of computing the learning objective is independent of the number of objects in the scene.

We translate this idea into a neural fusion field framework, which we call *Contrastive Lift*. We build on the recent progress in self-supervised learning, and combine two key ideas: the usage of a

contrastive loss, and the usage of a slow-fast learning scheme for minimizing the latter in a stable manner. We believe to be the first to introduce these two ideas in the context of neural fields.

We compare our method to recent techniques including Panoptic Lifting [48] on standard 3D instance segmentation benchmarks, *viz.* ScanNet [13], Replica [50], and Hypersim [46]. To better demonstrate the scalability of our method to a very large number of object instances, we introduce a semi-realistic Messy Rooms dataset featuring scenes with up to 500 objects.

## 2    Related Work

**Neural Radiance Fields (NeRFs).** NeRF [40] and its numerous variants [2, 5, 34, 36, 42] have achieved breakthrough results in generating photorealistic 3D reconstructions from 2D images of a scene. These systems typically represent the scene as a continuous volumetric function that can be evaluated at any 3D point, enabling high-quality rendering of novel views from any viewpoint.

**Objects and Semantics in NeRF.** While NeRF by default offers low-level modelling of radiance and geometry, recent methods have expanded the set of tasks that can be addressed in this context to include *semantic* 3D modelling and scene decomposition. Some works use neural scene representations to decompose scenes into foreground and background without supervision or from weak signals [17, 41, 47, 56, 62, 63], such as text or object motion. Others exploit readily available annotations for 2D datasets to further extend the capabilities of NeRF models. For example, Semantic NeRF [67] proposes to incorporate a separate branch predicting semantic labels, while NeSF [60] predicts a semantic field by feeding a density field as input to a 3D semantic segmentation model.

Closer to our work are methods that employ NeRFs to address the problem of 3D panoptic segmentation [19, 26, 31, 48, 61]. Panoptic NeRF [19] and Instance-NeRF [26] make use of 3D instance supervision. In Panoptic Neural Fields [31], each instance is represented with its own MLP but dynamic object tracking is required prior to training the neural field. In this work, we focus on the problem of lifting 2D instance segmentation to 3D without requiring any 3D masks or object tracks. A paper most related to our work is Panoptic Lifting [48], which also seeks to solve the same problem, using linear assignment to make multi-view annotations consistent. Here, we propose a more efficient and effective technique based on learning permutation-invariant embedding vectors instead.

**Fusion with NeRF.** The aforementioned works, such as Semantic NeRF [67] or Panoptic Lifting [48], are also representative of a recent research direction that seeks to *fuse* the output of 2D analysis into 3D space. This is not a new idea; multi-view semantic fusion methods [25, 35, 37, 38, 51, 59] predate and extend beyond NeRF. The main idea is that multiple 2D semantic observations (*e.g.*, noisy or partial) can be combined in 3D space and re-rendered to obtain clean and multi-view consistent labels. Instead of assuming a 3D model, others reconstruct a semantic map incrementally using SLAM [32, 43, 53]. Neural fields have greatly improved the potential of this idea. Instead of 2D labels, recent works, such as FFD [29], N3F [55], and LERF [27], apply the 3D fusion idea directly to supervised and unsupervised dense features; in this manner, unsupervised semantics can be transferred to 3D space, with benefits such as zero-shot 3D segmentation.

**Slow-fast contrastive learning.** Many self-supervised learning methods are based on the idea of learning representations that distinguish different samples, but are similar for different augmentations of the same sample. Some techniques build on InfoNCE [54, 57] and, like MoCo [23] and SimCLR [7], use a contrastive objective. Others such as SWaV [3] and DINO [4] are based on online pseudo-labelling. Many of these methods stabilise training by using mean-teachers [52], also called momentum encoders [23]. The idea is to have two versions of the same network: a *fast* "student" network supervised by pseudo-labels generated from a *slow* "teacher" network, which is in turn updated as the moving average of the student model. Our formulation is inspired by this idea and extends it to learning neural fields.

**Clustering operators for segmentation.** Some works [14, 18, 30, 45] have explored using clustering of pixel-level embeddings to obtain instance segment assignments. Recent works [64, 65] learn a pixel-cluster assignment by reformulating cross-attention from a clustering perspective. Our proposed method, Contrastive Lift, is similar in spirit, although we learn the embeddings (and cluster centers) using volumetric rendering from 2D labels.

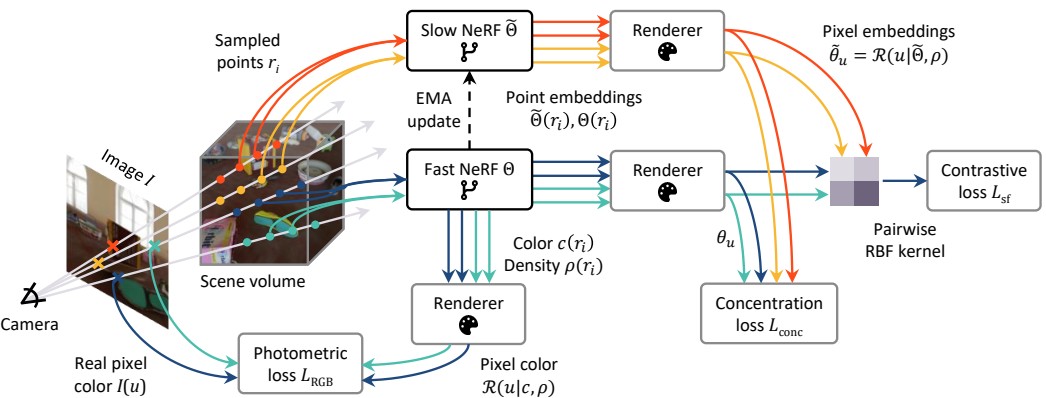

Figure 2: Overview of the Contrastive Lift architecture. See Section 3 for details.

# 3 Proposed Method: Contrastive Lift

Here and in Fig. 2, we describe Contrastive Lift, our approach for fusing 2D instance segmentation in 3D space. An image is a mapping $I : \Omega \to \mathbb{R}^3$, where $\Omega$ is a pixel grid in $\mathbb{R}^2$, and the values are RGB colours. We have a set of images $\mathcal{I}$ captured in the same scene and, for each image $I \in \mathcal{I}$, we have its camera pose $\pi \in SE(3)$ as well as object identity labels $y : \Omega \to \{1, \dots, L\}$ obtained from a 2D instance segmentation model for the image $I$. The labels $y$ assigned to the 3D objects in one image $I$ and the labels $y'$ in another image $I'$ are in general not consistent. Furthermore, these 2D label maps can be noisy across views.

We use this data to fit a neural field. The latter is a neural network that maps 3D coordinates $x \in \mathbb{R}^3$ to multiple quantities. The first two quantities are density, denoted by $\rho : \mathbb{R}^3 \mapsto [0, 1]$, and radiance (colour), denoted by $c : \mathbb{R}^3 \times \mathbb{S}^2 \mapsto [0, 1]^3$. Following the standard neural radiance field approach [36], the colour $c(x, d)$ also depends on the viewing direction $d \in \mathbb{S}^2$. The third quantity is a $D$-dimensional instance embedding (vector) denoted as $\Theta : \mathbb{R}^3 \mapsto \mathbb{R}^D$. Each 3D coordinate is also mapped to a semantic embedding that represents a distribution over the semantic classes.

**Differentiable rendering.** The neural field associates attributes (density, colour, and embedding vectors) to each 3D point $x \in \mathbb{R}^3$. These attributes are projected onto an image $I$ taken from a viewpoint $\pi$ via differentiable ray casting. Given a pixel location $u \in \Omega$ in the image, we take $N$ successive 3D samples $r_i \in \mathbb{R}^3$, $i = 0, \dots, N-1$ along the ray from the camera center through the pixel (so that $(u, f) \propto \pi^{-1}(r_i)$ where $f$ is the focal length). The probability that a photon is not absorbed when travelling from sample $r_i$ to sample $r_{i+1}$ is $\exp(-\rho(r_i)\delta_i)$ where $\delta_i = \|r_{i+1} - r_i\|_2$ is the distance between points. The *transmittance* $\tau_i = \exp(-\sum_{j=0}^{i-1} \rho(r_j)\delta_j)$ is the probability that the photon travels through sample $r_i$. The projection of any neural field $\mathbf{f}$ onto pixel $u$ is thus given by the rendering equation:

$$\mathcal{R}(u|\mathbf{f}, \rho, \pi) = \sum_{i=0}^{N-1} \mathbf{f}(r_i)(\tau_i - \tau_{i+1}) = \sum_{i=0}^{N-1} \mathbf{f}(r_i)\tau_i(1 - \exp(-\rho(r_i)\delta_i)) \tag{1}$$

In particular, the colour of a pixel is reconstructed as $I(u) \approx \mathcal{R}(u|c(\cdot, d_u), \rho, \pi)$ where the viewing direction $d_u = r_0/\|r_0\|_2$. The photometric loss is thus:

$$\mathcal{L}_{\text{RGB}}(c, \rho|I) = \frac{1}{|\Omega|} \sum_{u \in \Omega} \|I(u) - \mathcal{R}(u|c(\cdot, d_u), \rho, \pi)\|^2. \tag{2}$$

**Instance embeddings and slow-fast contrastive learning.** The photometric loss (2) learns the colour and density fields $(c, \rho)$ from the available 2D views $\mathcal{I}$. Now we turn to learning the instance embedding field $\Theta : \mathbb{R}^3 \mapsto \mathbb{R}^D$. As noted in Section 1, the goal of the embeddings is to capture the (binary) distances between pixel labels sufficiently well. By that, we mean that the segments can be recovered, modulo a permutation of their labels, by simply *clustering* the embeddings a posteriori.

We cast learning the embeddings as optimising the following contrastive loss function:

$$\mathcal{L}_{\text{contr}}(\Theta, \rho | y) = -\frac{1}{|\Omega|} \sum_{u \in \Omega} \log \frac{\sum_{u' \in \Omega} \mathbf{1}_{y(u)=y(u')} \exp(\text{sim}(\theta_u, \theta_{u'}; \gamma))}{\sum_{u' \in \Omega} \exp(\text{sim}(\theta_u, \theta_{u'}; \gamma))}, \quad \theta_u = \mathcal{R}(u | \Theta, \rho, \pi),$$
(3)

where $\mathbf{1}$ is the indicator function, and $\text{sim}(x, x'; \gamma) = \exp(-\gamma \|x - x'\|^2)$ is a Gaussian RBF kernel used to compute the similarity between embeddings in Euclidean space. Therefore, pixels that belong to the same segment are considered positive pairs, and their embeddings are brought closer, while the embeddings of pixels from different segments are pushed apart. It is worth emphasizing that, since the object identity labels obtained from the underlying 2D segmenter are not consistent *across* images, $\mathcal{L}_{\text{contr}}$ is only applied to positive and negative pixel pairs sampled from the *same* image.

While Eq. (3) is logically sound, we found it to result in gradients with high variance. To address this, we draw inspiration from momentum-teacher approaches [1, 4, 24] and define a *slowly-updated* instance embedding field $\tilde{\Theta}$, with parameters that are updated with an exponential moving average of the parameters of $\Theta$, instead of gradient descent. With this, we reformulate Eq. (3) as:

$$\mathcal{L}_{\text{sf}}(\Theta, \rho | y, \tilde{\Theta}) = -\frac{1}{|\Omega_1|} \sum_{u \in \Omega_1} \log \frac{\sum_{u' \in \Omega_2} \mathbf{1}_{y(u)=y(u')} \exp(\text{sim}(\theta_u, \tilde{\theta}_{u'}; \gamma))}{\sum_{u' \in \Omega_2} \exp(\text{sim}(\theta_u, \tilde{\theta}_{u'}; \gamma))},$$
(4)

where $\theta_u = \mathcal{R}(u | \Theta, \rho, \pi)$, and $\tilde{\theta}_{u'} = \mathcal{R}(u' | \tilde{\Theta}, \rho, \pi)$. Here, we randomly partition the pixels $\Omega$ into two non-overlapping sets $\Omega_1$ and $\Omega_2$, one for the "fast" embedding field $\Theta$, and another for the "slow" field $\tilde{\Theta}$. This avoids the additional cost of predicting and rendering each pixel's embedding using both models, and allows the computational cost to remain the same as for Eq. (3).

**Concentration loss.** In order to further encourage the separation of the embedding vectors $\Theta$ and thus simplify the extraction of the objects via *a posteriori* clustering, we introduce a loss function that further encourages the embeddings to form concentrated clusters for each object:

$$\mathcal{L}_{\text{conc}}(\Theta, \rho | y, \tilde{\Theta}) = \frac{1}{|\Omega_1|} \sum_{u \in \Omega_1} \left\| \theta_u - \frac{\sum_{u' \in \Omega_2} \mathbf{1}_{y(u)=y(u')} \tilde{\theta}_{u'}}{\sum_{u' \in \Omega_2} \mathbf{1}_{y(u)=y(u')}} \right\|^2.$$
(5)

This loss computes a centroid (average) embedding as predicted by the "slow" field $\tilde{\Theta}$ and penalizes the squared error between each embedding (as predicted by the "fast" field $\Theta$) and the corresponding centroid. While this loss reduces the variance of the clusters, it is not a sufficient training objective by itself as it does not encourage the separation of different clusters, as done by Eqs. (3) and (4).

**Semantic segmentation.** For semantic segmentation, we follow the same approach as Semantic NeRF [67], learning additional embedding dimensions (one per semantic class), rendering labels in the same manner as Eq. (1), and using the cross-entropy loss for fitting the semantic field. Additionally, we also leverage the segment consistency loss introduced in [48] which encourages the predicted semantic classes to be consistent within an image segment.

**Architectural details.** Our neural field architecture is based on TensoRF [6]. For the density, we use a single-channel grid whose values represent the scalar density field directly. For the colour, a multi-channel grid predicts an intermediate feature which is concatenated with the viewing direction and passed to a shallow 3-layer MLP to predict the radiance field. The viewing directions are encoded using a frequency encoding [40, 58]. For the instance embedding field $\Theta$ (and also the "slow" field $\tilde{\Theta}$ which has the exact same architecture as the "fast" field $\Theta$), we use a shallow 5-layer MLP that predicts an embedding given an input 3D coordinate. The same architecture is used for the semantic field. We use raw 3D coordinates directly *without* a frequency encoding for the instance and semantic components. More details are provided in the supplementary material.

**Rendering instance segmentation maps.** After training is complete, we sample $10^5$ pixels from 100 random viewpoints (not necessarily training views) and render the *fast* instance field $\Theta$ at these pixels using the corresponding viewpoint pose. The rendered $10^5 \times D$ embeddings are clustered using HDBSCAN [39] to obtain centroids, which are cached. Now, for any novel view, the field $\Theta$ is rendered and for each pixel, the label of the centroid nearest to the rendered embedding is assigned.

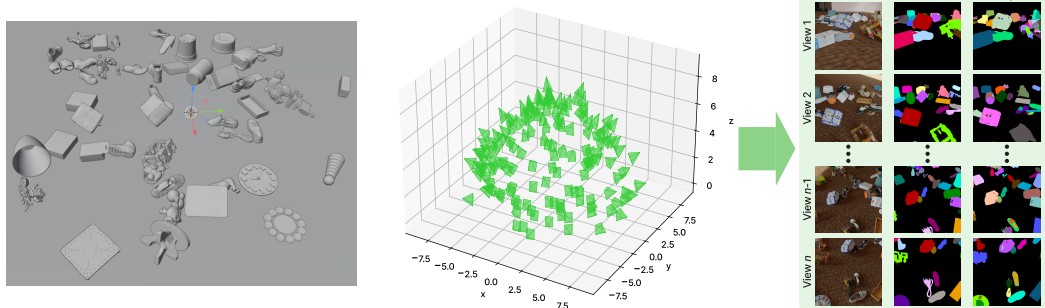

Figure 3: Messy Rooms dataset visualization. Left: physically realistic static 3D scene with $N$ objects from GSO [15]. Middle: $M$ camera viewpoints sampled in a dome-shaped shell. Right: ground-truth RGB and instance IDs, and instance segmentations obtained from Detic [68].

## 4 Messy Rooms Dataset

In order to study the scalability of our method to scenes with a large number of objects, we generate a semi-realistic dataset using Kubric [20]. To generate a scene, we first spawn $N$ realistically textured objects, randomly sampled from the Google Scanned Objects dataset [15], without any overlap. The objects are dropped from their spawned locations and a physics simulation is run for a few seconds until the objects settle in a natural arrangement. The static scene is rendered from $M$ inward-facing camera viewpoints randomly sampled in a dome-shaped shell around the scene. Background, floor, and lighting are based on 360° HDRI textures from PolyHaven [66] projected onto a dome.

Specifically, we create scenes with $N = 25, 50, 100$, and $500$ objects. The number of viewpoints, $M$ is set to $\min(1200, \lfloor 600 \times \sqrt{N/25} \rfloor)$, and the rendered image resolution is $512 \times 512$. To ensure that the focus is on the added objects, we use background textures *old_room* and *large_corridor* from PolyHaven that do not contain any objects. A total of 8 scenes are generated. The use of realistic textures for objects and background environments makes them representative of real-world scenarios.

Additionally, we would like to maintain a consistent number of objects per image as we increase the total number of objects so that the performance of the 2D segmenter is not a factor in the final performance. Firstly, we ensure that the floor area of the scene scales proportionally with the number of objects, preventing objects from becoming densely packed. Secondly, the cameras move further away from the scene as its extent increases. To ensure that the same number of objects is visible in each image, regardless of the scene size, we adjust the focal length of the cameras accordingly, *i.e.*, $f = 35.0 \times \sqrt{N/25}$, creating an effect similar to magnification. This approach ensures a comparable object distribution in each image, while enabling us to study the scalability of our method.

We render the instance IDs from each camera viewpoint to create ground-truth instance maps. These ground-truth instance IDs remain consistent (tracked) across views, as they are rendered from the same 3D scene representation.[1] Figure 3 shows illustrative examples from the dataset, which we name *Messy Rooms*. For evaluation (Section 5), semantic maps are required. As there is a large variety of different object types in Kubric, there is no off-the-shelf detector that can classify all of these, and since we are interested in the instance segmentation problem, rather than the semantic classes, we simply lump all object types in a single "foreground" class, which focuses the evaluation on the quality of instance segmentation. More details about the dataset are provided in Appendix A.

## 5 Experiments

**Benchmarks and baselines.** We train and evaluate our proposed method on challenging scenes from the ScanNet [13], Hypersim [46], and Replica [50] datasets. We compare our method with Panoptic Lifting (PanopLi) [48], which is the current state-of-the-art for lifting 2D panoptic predictions to 3D, along with other 3D panoptic segmentation approaches: Panoptic Neural Fields [31] and

---

[1]In all experiments, *tracked* ground-truth instance maps are used only for evaluation and not to train models.

Table 1: Results on ScanNet, Hypersim, and Replica datasets. The performance of all prior work has been sourced from [48]. For each dataset, we report the PQ$^{\text{scene}}$ metric.

| Method | ScanNet [13] | HyperSim [46] | Replica [50] |
|---|---|---|---|
| DM-NeRF [61] | 41.7 | 51.6 | 44.1 |
| PNF [31] | 48.3 | 44.8 | 41.1 |
| PNF + GT BBoxes | 54.3 | 47.6 | 52.5 |
| PanopLi [48] | 58.9 | 60.1 | 57.9 |
| Vanilla (**Ours**) | 60.5 | 60.9 | 57.8 |
| Slow-Fast (**Ours**) | **62.3** | **62.3** | **59.1** |

Table 2: Results on the Messy Rooms dataset. PQ$^{\text{scene}}$ metric is reported on "old room" and "large corridor" environments with increasing number of objects in the scene ($N = 25, 50, 100, 500$).

| Method | Old Room environment | | | | Large Corridor environment | | | |
|---|---|---|---|---|---|---|---|---|
| | 25 Objects | 50 Objects | 100 Objects | 500 Objects | 25 Objects | 50 Objects | 100 Objects | 500 Objects |
| PanopLi [48] | 73.2 | 69.9 | 64.3 | 51.0 | 65.5 | 71.0 | 61.8 | 49.0 |
| Vanilla (**Ours**) | 74.1 | 71.2 | 63.6 | 49.7 | 67.9 | 69.3 | 62.2 | 47.2 |
| Slow-Fast (**Ours**) | **78.9** | **75.8** | **69.1** | **55.0** | **76.5** | **75.5** | **68.7** | **52.5** |

DM-NeRF [61]. We follow PanopLi [48] for the data preprocessing steps and train-test splits for each scene from these datasets. We also evaluate our proposed method and PanopLi on our Messy Rooms dataset (Section 4) that features scenes with up to 500 objects. These experiments aim to demonstrate the scalability of our proposed method as compared to the linear-assignment approach.

We compare two variants of our Contrastive Lift method: (1) *Vanilla*: uses the simple contrastive loss (Eq. (3)), and (2) *Slow-Fast*: uses slow-fast contrastive (Eq. (4)) and concentration (Eq. (5)) losses.

**Metrics.** The metric used in our evaluations is the scene-level Panoptic Quality (PQ$^{\text{scene}}$) metric introduced in [48]. PQ$^{\text{scene}}$ is a scene-level extension of standard PQ [28] that takes into account the consistency of instance IDs across views/frames (*aka* tracking). In PQ$^{\text{scene}}$, predicted/ground-truth segments with the same instance ID across all views are merged into *subsets* and all pairs of predicted/ground-truth *subsets* are compared, marking them as a match if the IoU is greater than 0.5.

**Implementation Details.** We train our neural field model for 400k iterations on all scenes. Optimization-related hyper-parameters can be found in Appendix B.2. The density grid is optimised using only the photometric loss ($\mathcal{L}_{\text{RGB}}$). While rendering the instance/semantic fields and computing associated losses (Eqs. (3) to (5)), gradients are stopped from flowing to the density grid.

For experiments on ScanNet, Hypersim and Replica, we use Mask2Former (M2F) [8] as the 2D segmenter to obtain the image-level semantic labels and instance identities. Although any 2D segmenter can be used, using M2F allows direct comparisons with other state-of-the-art approaches [48]. We follow the protocol used in [48] to map the COCO [33] vocabulary to 21 classes in ScanNet.

For experiments on Messy Rooms, we use Detic [68] instead since the object categories are not isomorphic to the COCO vocabulary M2F uses. We use the LVIS [21] vocabulary with Detic. To show the scalability of our method compared to a linear-assignment-based approach, we train the PanopLi [48] model on this dataset. For fair comparison, we first train the density, colour and semantic fields, which are identical in PanopLi and our approach. We then separately train the instance field using the respective linear-assignment and slow-fast contrastive losses, with all other components frozen, ensuring that performance is only influenced by the quality of the learned instance field.

## 5.1 Results

In Table 1, we compare the performance of our proposed approach with existing methods on three datasets: ScanNet [13], HyperSim [46], and Replica [50]. Since the semantic field and underlying TensoRF [5] architecture we use is similar to Semantic-NeRF [67] and PanopLi [48], we only report the PQ$^{\text{scene}}$ metric here and have added an additional table to Appendix D where we show that the mIoU and PSNR of our method match the performance of prior methods as expected. We observe that the proposed *Slow-Fast* approach consistently outperforms the baselines on all three datasets, while also outperforming the state-of-the-art Panoptic Lifting [48] method by $+3.9$, $+1.4$ and $+0.8$

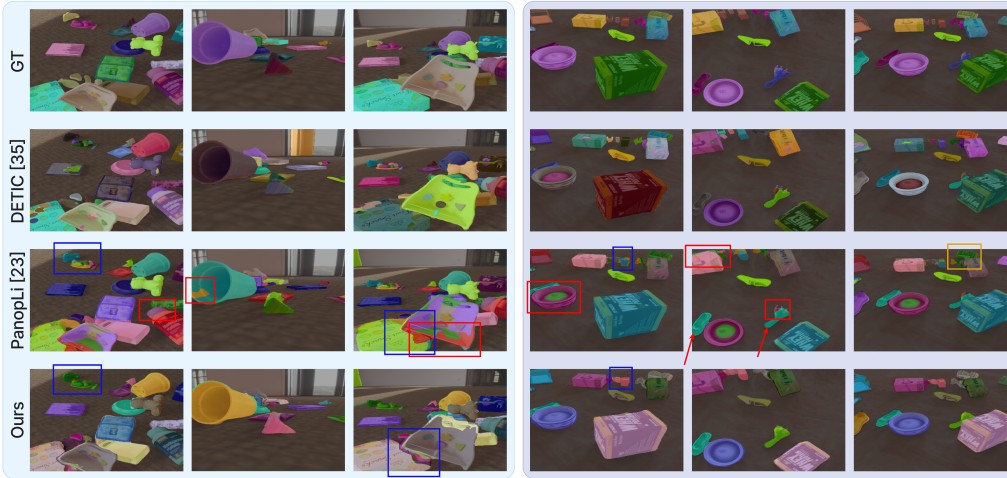

(a) Messy Rooms: `large_corridor` (25 objects)    (b) Messy Rooms: `old_room` (25 objects)

Figure 4: Qualitative comparisons of our method with PanopLi [48] and Detic [68] (underlying 2D segmenter model) on scenes from our Messy Rooms dataset. **Colour coding:** regions where PanopLi performs poorly are highlighted with **red** boxes, while regions where both PanopLi and our method exhibit poor performance are marked with **blue** boxes. Additionally, **red** arrows indicate instances where PanopLi fails to distinguish between different objects. Please zoom in to observe finer details.

Table 3: Ablations of different variants of the Contrastive Lift method. $PQ^{scene}$ metric averaged over the scenes of ScanNet and Messy Rooms datasets is reported. Embedding size of 3 is used.

| Dataset | $\mathcal{L}_{sf}+\mathcal{L}_{conc}$ | $\mathcal{L}_{sf}$ | $\mathcal{L}_{contr}$ | $\mathcal{L}_{contr}+\mathcal{L}_{conc}$ (fast) |
|---|---|---|---|---|
| ScanNet [13] | **62.0** | 61.3 | 60.5 | 55.2 |
| Messy Rooms | **69.0** | 66.5 | 63.2 | 51.7 |

$PQ^{scene}$ points on these datasets respectively. We note that the *Vanilla* version of our method also performs comparably with PanopLi and outperforms other methods on all datasets.

Table 2 shows comparisons between our method and PanopLi [48] on scenes from our Messy Rooms dataset with $25, 50, 100,$ and $500$ objects. We see that the margin of improvement achieved by Contrastive Lift over PanopLi is even larger on these scenes, which shows that the proposed method scales favorably to scenes with a large number of objects. Fig. 4 shows qualitative results on two of these scenes. Even though the 2D segments obtained using Detic [68] are noisy (sometimes *over-segmented*) and generally labelled inconsistently between views, the resulting instance segmentations rendered by Contrastive Lift are clearer and consistent across views. We also note that PanopLi sometimes fails to distinguish between distinct objects as pointed out in Fig. 4b.

## 5.2 Ablations

**Different variants of Contrastive Lift.** Our proposed method uses $\mathcal{L}_{sf}$ (Eq. (4)) and $\mathcal{L}_{conc}$ (Eq. (5)) to optimise the instance embedding field. To study the effect of these losses, we design a comprehensive set of variants of the proposed method: **(1)** Proposed ($\underline{\mathcal{L}_{sf}+\mathcal{L}_{conc}}$), **(2)** Proposed without Concentration loss ($\underline{\mathcal{L}_{sf}}$), **(3)** Vanilla contrastive ($\underline{\mathcal{L}_{contr}}$), **(4)** Vanilla contrastive with Concentration loss applied to "fast" field since there is no "slow" field ($\underline{\mathcal{L}_{contr}+\mathcal{L}_{conc}(\text{fast})}$). Table 3 shows these ablations.

**Effect of embedding size on performance.** We investigate the impact of varying the instance embedding size on the performance of our proposed Contrastive Lift method. Specifically, we evaluate the effect of different embedding sizes using the $PQ^{scene}$ metric on ScanNet, Hypersim and Replica datasets. As shown in Fig. 5, we find that an embedding size as small as 3 is already almost optimal. Based on this, we use an embedding size of $24$ for experiments with these datasets (*c.f.* Table 1). For experiments with Messy Rooms dataset (*c.f.* Table 2), we keep the embedding size to 3.

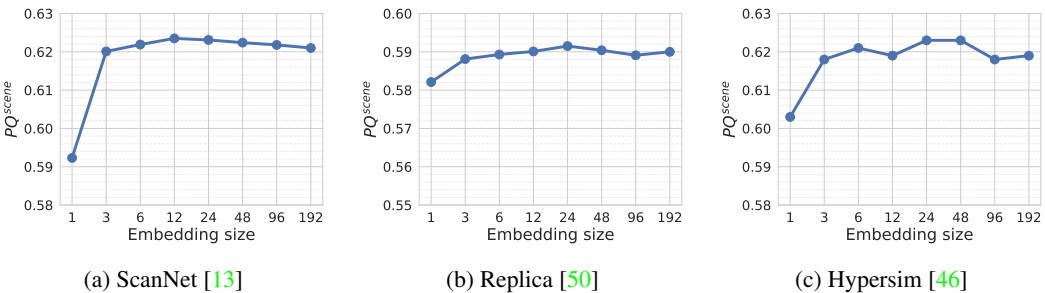

(a) ScanNet [13]      (b) Replica [50]      (c) Hypersim [46]

Figure 5: Impact of the embedding size on the performance (PQ^scene) of the instance module.

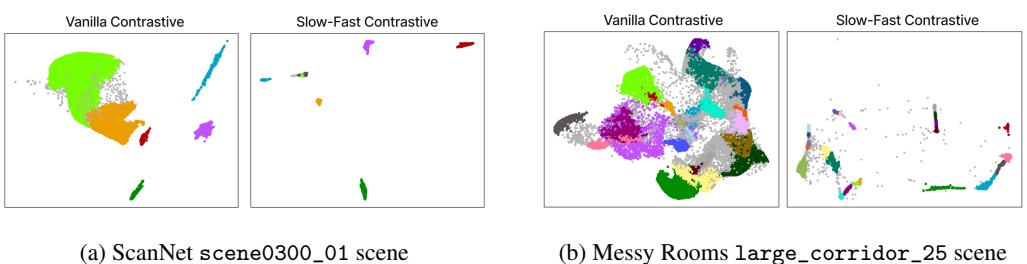

(a) ScanNet `scene0300_01` scene    (b) Messy Rooms `large_corridor_25` scene

Figure 6: Embeddings obtained using vanilla (plain) contrastive learning and our proposed Slow-Fast contrastive learning. We use LDA [22] to project the embeddings to 2D for the illustration here.

**Qualitative evaluation: Slow-fast vs vanilla contrastive learning.** Fig. 6 shows how the embeddings are distributed in Euclidean space when learned using our proposed slow-fast contrastive loss (Eqs. (4) and (5)) and the vanilla contrastive loss (Eq. (3)). Embeddings learned with the slow-fast method are clustered more compactly and are easy to distinguish using any post-processing algorithm, such as HDBSCAN [39] which is used in this example.

**Comparison to underlying 2D instance segmentation model with tracking.** Before lifting, the predictions of the underlying 2D instance segmentation model (e.g., Mask2Former [8] or Detic [68]) are not consistent (*aka* tracked) across frames/views. To achieve consistency and to allow comparisons with our approach, we post-process the 2D segmenter's predictions using Hungarian Matching for cross-frame tracking as follows:

1. **w/ Hungarian matching (2D IoU)**: Given sets of predicted segments ($P_i$ and $P_{i+1}$) from consecutive frames, compute IoU matrix by comparing all segment pairs in $P_i \times P_{i+1}$. Apply Hungarian matching to the IoU matrix to associate instance segments across frames.

2. **w/ Hungarian matching based on IoU after depth-aware pose-warping**: Use ground-truth pose and depth for warping $(i+1)$-th frame's segmentation to frame $i$. Compute IoU matrix using warped segmentations and apply Hungarian matching.

3. **w/ Hungarian matching using ground-truth pointcloud**: Using only consecutive frames leads to errors in long-range tracking. To address this, starting from the first frame, unproject 2D segments into the 3D point cloud. Iteratively fuse segments in 3D using Hungarian matching. This way, segments from preceding frames along with 3D information are used for tracking.

The last two baselines use 3D groundtruth for tracking. Table 4 shows that despite 3D information being used for matching, Contrastive Lift still significantly improves over the underlying 2D model.

**Frame-level improvement on underlying 2D segmentation models.** In addition to generating consistent (tracked) instance segmentations, our method also improves the per-frame quality (i.e., not considering tracking) of the underlying 2D segmentation model. To show this, we train Contrastive Lift on ScanNet scenes with different 2D models, *viz.* Mask2Former [8], MaskFormer [9] and Detic [68]. In Table 5 we report the Panoptic Quality (PQ) metric (computed per frame) for these 2D models and for our method when trained with segments from each corresponding model.

Table 4: Comparison of our approach with the underlying 2D segmentations on ScanNet [13]. For M2F predictions [8], consistency across frames is obtained with different tracking variants.

| Method | PQ$^{scene}$ |
|---|---|
| Mask2Former [8] (M2F) (**non-tracked**) | 32.3 |
| M2F w/ Tracking method (1) | 33.7 |
| M2F w/ Tracking method (2) | 34.0 |
| M2F w/ Tracking method (3) | 41.0 |
| Contrastive Lift (**ours** trained w/ Mask2Former labels) | **62.3** |

Table 5: Improvement of per-frame segmentation quality as measured by Panoptic Quality (PQ).

| Method | PQ |
|---|---|
| MaskFormer [9] | 41.1 |
| Contrastive Lift (w/ MaskFormer labels) | 61.7 |
| Mask2Former [8] | 42.0 |
| Contrastive Lift (w/ Mask2Former labels) | 61.6 |
| Detic [68] | 43.6 |
| Contrastive Lift (w/ Detic labels) | **62.1** |

**Comparison of training speed with the linear-assignment loss method.** While the exact number of objects present in a scene is unknown, linear assignment-based methods typically require a hyperparameter $K$ that specifies the *maximum* number of objects. Solving the linear assignment problem in PanopLi's loss is $O(K^3)$ [48]. Our method is agnostic to object count, eliminating the need for such a parameter. Our approach does rely on the size of the embedding size, but, as shown above, even a very small size suffices. In the slow-fast contrastive loss computation, the Softmax function dominates more than the pairwise similarity matrix calculation. Consequently, we find that the training speed of Contrastive Lift is largely unaffected by the choice of embedding size.

Table 6 compares the training speed, measured on a NVIDIA A40 GPU, between PanopLi and our method, showing that PanopLi iterations become slower as $K$ increases. We only optimise the instance embedding field with associated losses, while the density/colour/semantic fields are frozen.

Table 6: Training speed in iterations/second. Mean $\pm$ error margin measured over 8 runs.

| Contrastive Lift | Panoptic Lifting [48] | | | |
|---|---|---|---|---|
| | $K = 25$ | $K = 50$ | $K = 100$ | $K = 500$ |
| **16.06 $\pm$ 2.34** | 13.01 $\pm$ 1.26 | 12.53 $\pm$ 0.92 | 12.10 $\pm$ 1.07 | 9.41 $\pm$ 0.60 |

# 6 Limitations

Contrastive Lift improves noisy 2D input segmentations, but cannot recover from catastrophic failures, such as entirely missing object classes. It also requires the 3D reconstruction to work reliably. As a result, we have focused on static scenes, as 3D reconstruction remains unreliable in a dynamic setting. Contrastive Lift is a useful building block in applications, but has no particular direct societal impact. The datasets used in this paper are explicitly licensed for research and contain no personal data.

# 7 Conclusion

We have introduced Contrastive Lift, a method for fusing the outputs of 2D instance segmenter using a 3D neural fields. It learns a 3D vector field that characterises the different object instances in the scene. This field is fitted to the output of the 2D segmenter in a manner which is invariant to permutation of the object labels, which are assigned independently and arbitrarily in each input image. Compared to alternative approaches that explicitly seek to make multi-view labels compatible, Contrastive Lift is more accurate and scalable, enabling future work on larger object collections.

## Acknowledgments and Disclosure of Funding

We are grateful for funding from EPSRC AIMS CDT EP/S024050/1 and AWS (Y. Bhalgat), ERC-CoG UNION 101001212 (A. Vedaldi and I. Laina), EPSRC VisualAI EP/T028572/1 (I. Laina, A. Vedaldi and A. Zisserman), and Royal Academy of Engineering RF\201819\18\163 (J. Henriques).

**Data Ethics.** We use the Google Scanned Objects, ScanNet, Hypersim and Replica datasets following their terms and conditions. These datasets do not contain personal data. For further details on ethics, data protection, and copyright please see https://www.robots.ox.ac.uk/~vedaldi/research/union/ethics.html.

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

## A  Messy Rooms dataset

The full Messy Rooms dataset introduced in this work can be accessed at this link: [https://figshare.com/s/b195ce8bd8eafe79762b](https://figshare.com/s/b195ce8bd8eafe79762b). We show some representative examples from this dataset below in Fig. 14, 15, 16, and 17, which illustrate scenes with $25, 50, 100$ and $500$ objects respectively. Notice how the density of "number of objects per image" remains similar as the number of objects increases from $25$ to $500$. In Fig. 18, we show the corresponding 3D scenes used to generate the datasets.

## B  Implementation Details

Here, we provide further implementation details for our method in addition to the details mentioned in Sections 3 and 5 of the main paper.

### B.1  Architectural details

Our neural field architecture is similar to [48] for fairness of comparisons. The density and color grids are initialized with a resolution of $128 \times 128 \times 128$ which is progressively increased up to $192 \times 192 \times 192$ by the end of training. The density and color grids use $16$ and $48$ components respectively. The output of the color grid is projected to $27$ dimensions which are then processed by a 3-layer MLP with $128$ hidden units per layer to output the RGB color. The *fast* and *slow* instance fields use a $256$ hidden size in their MLP, while the semantic field uses a hidden size of $128$.

### B.2  Training details

We follow a schedule for training our neural field model as follows: (1) For the first 40k iterations, the model is trained only with the RGB reconstruction loss ($\mathcal{L}_{\text{RGB}}$). In this initial phase, the density field is optimized to reach a reasonable quality such that it can be used to render the instance/semantic field. (2) At 40k iterations, the semantic segmentation loss (i.e., cross-entropy loss, as in [48, 67]) is activated and used for the rest of the training iterations. (3) At 160k iterations, the instance embedding loss (i.e., $\mathcal{L}_{\text{sf}} + \mathcal{L}_{\text{conc}}$ for the *slow-fast* version of our method or $\mathcal{L}_{\text{contr}}$ for the *vanilla* baseline) is activated. (4) At 280k iterations, the segment consistency loss (proposed in [48]) is activated. For scenes from the Hypersim dataset [46], we activate the segment consistency loss at 200k iterations instead. In our proposed slow-fast clustering framework, the *slow* field parameters are updated using an exponential moving average with momentum $m = 0.9$, i.e. $\tilde{\Theta} = \tilde{\Theta} \times m + \Theta \times (1 - m)$.

The RGB reconstruction loss, semantic segmentation loss, instance embedding loss, and segment consistency loss are balanced using weights of $1.0, 0.1, 0.1$, and $1.0$ respectively. However, we empirically observe that the final performance is not very sensitive to these choices. A learning rate of $5 \cdot 10^{-4}$ is used for all MLPs and $0.01$ for the grids. A batch-size of $2048$ is used to train all models.

### B.3  Post-processing Clustering details

Given the learned instance embedding field, a clustering mechanism (e.g., HDBSCAN [39]) can be used to obtain cluster centroids and generate instance segmentation maps. We have chosen HDBSCAN since it does not require the number of objects to be known *a priori*. Generally, clusters obtained by HDBSCAN are non-convex and assigning the label of the nearest centroid is not recommended. But, our proposed method results in highly compact clusters which makes this simple method effective. We perform clustering as follows.

**Hierarchical clustering using semantic predictions.** An advantage of our method is that we can use our model's semantic predictions to guide the clustering of the instance embeddings. "Instance" segmentation requires separating instances of the same semantic class. Based on this, we perform hierarchical clustering as follows:

1. After training, sample $10^5$ pixels from 100 random viewpoints and render the fast instance field $\Theta$ and semantic field for these pixels.
2. Group the $10^5$ rendered embeddings based on predicted semantic labels, forming $S$ groups.

3. Cluster the embeddings within each group separately using HDBSCAN, caching cluster-centroids for each group, assigning a unique instance label to each centroid.

4. For a novel view, render the instance field and semantic field, assigning each pixel an instance embedding and semantic class. Obtain the instance label for a pixel by finding the closest centroid to the rendered instance embedding within the group of the same semantic label as the pixel.

**Tuning clustering hyperparameter.** Despite HDBSCAN's robustness to hyperparameter selection, we found that it is beneficial to specify a *minimum cluster size*. Since we always sample and render $10^5$ pixels for clustering, the expected cluster size per object decreases as the number of objects increases. To determine an optimal value, we perform a hyperparameter sweep using $10\%$ of the training data, which includes training viewpoints and associated segments from the 2D segmenter. We then use this identified optimal value to perform clustering as described above.

## C  Comparison between different clustering algorithms.

We compare HDBSCAN with other unsupervised clustering algorithms, *viz.* MeanShift [11] and DBSCAN [16]. We tune the *bandwidth* parameter with MeanShift, and the *epsilon* parameter with DBSCAN. However, we note that MeanShift struggles to converge for embedding sizes greater than 10. For fair comparison, we train our model with an embedding size of 3. Table 7 show that both MeanShift and DBSCAN perform slightly worse but remain comparable to HDBSCAN. Generally, any unsupervised clustering method that doesn't require prior knowledge of the number of clusters is suitable for use with our method.

Table 7: Performance (PQ$^{\text{scene}}$) achieved with DBSCAN [16], MeanShift [11] and HDBSCAN [39].

|  | ScanNet [13] | Messy Rooms |
|---|---|---|
| w/ DBSCAN | 61.8 | 68.2 |
| w/ MeanShift | 62.0 | 68.6 |
| w/ HDBSCAN | 62.0 | 69.0 |

## D  Quality of our semantic and radiance field

In Tables 1 and 2 in the main paper, we evaluate the quality and consistency (*aka* tracking) of the instance segmentation maps obtained by the various tested methods. The semantic field and density/color field architecture of our method is based on Panoptic Lifting [48], which in turn is a modification of Semantic-NeRF [67] for the semantic component. As a sanity check, we compare the quality of rendered semantic and RGB maps obtained by these methods with ours. Table 8 shows the mean Intersection over Union (mIoU) and peak-signal-to-noise ratio (PSNR) metrics. As expected, the mIoU and PSNR obtained by our method is nearly the same as Panoptic Lifting.

For the Messy Rooms dataset we have explicitly ensured that the density and semantic model used by both Panoptic Lifting and our method are the same and the only factor influencing the final performance is the quality of the learned instance field. This is done by pre-training the same density and semantic fields for both methods and subsequently training the instance field using the respective objective functions.

Table 8: Comparisons of the rendered semantic and RGB maps. Performance numbers for [48, 67] are sourced from [48].

| Method | ScanNet [13] | | HyperSim [46] | | Replica [50] | |
|---|---|---|---|---|---|---|
| | mIoU | PSNR | mIoU | PSNR | mIoU | PSNR |
| Semantic-NeRF [67] | 58.9 | 26.6 | 58.5 | 24.8 | 59.2 | 26.6 |
| PanopLi [48] | **65.2** | **28.5** | 67.8 | **30.1** | **67.2** | 29.6 |
| Ours | **65.2** | 28.3 | **67.9** | 30.0 | 67.0 | 29.3 |

# E  Comparisons to other metric learning loss functions

While we employ a contrastive loss formulation to learn the instance embeddings, there are many alternative loss functions proposed in the metric learning literature. For comparison, we also train our instance embedding field with the Associative Embedding (AE) loss [44] and the margin-based contrastive loss [10].

To compute the AE Loss, we divide the batch $\Omega$ into groups based on segment ID. If there are $K$ groups/segments, $G_1 \ldots G_K$, then

$$\mathcal{L}_{\text{AE}}(\Theta, \rho | y) = \frac{1}{|\Omega|} \sum_k \sum_{u \in G_k} \|\theta_u - \bar{\theta}_k\|_2^2 + \frac{1}{K^2} \sum_k \sum_{k'} \|\bar{\theta}_k - \bar{\theta}_{k'}\|_2^2, \quad \bar{\theta}_k = \frac{1}{|G_k|} \sum_{u \in G_k} \theta_u \quad (6)$$

The margin-based contrastive loss is defined as:

$$\mathcal{L}_{\text{margin}}(\Theta, \rho | y) = \frac{1}{|\Omega|^2} \sum_{u, u' \in \Omega} \mathbf{1}_{[y(u) = y(u')]} \|\theta_u - \theta_{u'}\|_2^2 + \mathbf{1}_{[y(u) \neq y(u')]} \max(0, \epsilon - \|\theta_u - \theta_{u'}\|_2^2) \quad (7)$$

Here, $\theta_u = \mathcal{R}(u | \Theta, \rho, \pi)$ is the rendered instance field at pixel $u$. Note that, the slow-fast field formulation is not used in these comparisons. In Table 9 we compare the proposed objective (slow-fast) to these baselines. We observe that both the vanilla contrastive, as well as the slow-fast version of our method, outperform the alternatives.

**DINO-style loss.** Since our method is inspired by momentum-teacher approaches, e.g. DINO [4], we design a baseline with a DINO-style learning mechanism. Two pixels from the same instance segment are fed into the *slow* and *fast* fields. This is akin to DINO, where two random image transformations are fed to the student and teacher networks. A Centering layer is applied to the *slow* field embedding. A Projection module (with proj_dim = 512) is added to both the *slow* and *fast* fields followed by Softmax and a cross-entropy loss is used. After training, embeddings from the *fast* field are used for clustering. Results in Table 9 demonstrate that this baseline performs worse than the metric-learning losses on ScanNet. We do not evaluate this baseline on the Messy Rooms dataset.

Table 9: Comparing Associative Embedding Loss and Triplet Loss with our proposed losses (*vanilla* and *slow-fast*). An embedding size of 3 is used in all cases. PQ$^{\text{scene}}$ is reported here.

|  | ScanNet [13] | Messy Rooms |
|---|---|---|
| Panoptic Lifting [48] | 58.9 | 63.2 |
| Ours w/ AE loss ($\mathcal{L}_{\text{AE}}$) | 60.0 | 62.4 |
| Ours w/ Margin loss ($\mathcal{L}_{\text{margin}}$) | 60.1 | 62.9 |
| Ours w/ DINO-style loss | 54.7 | - |
| Ours w/ Vanilla contrastive loss ($\mathcal{L}_{\text{contr}}$) | 60.5 | 63.1 |
| Ours w/ Slow-Fast losses (**proposed**) | **62.0** | **69.0** |

# F  Stability of Slow-Fast loss compared to Vanilla contrastive loss

We found that training with the vanilla contrastive loss resulted in gradients with higher variance. The usage of a slowly-updated embedding field in the slow-fast loss formulation mitigates this problem and leads to more stable training. We quantitatively verify this by computing the *relative variance* (which is $Var(\cdot)/Mean(\cdot)$) in the gradients of the loss w.r.t. to the instance embeddings (*i.e.* $dL/d\Theta$). Figure 7 shows that the vanilla loss exhibits spikes with a maximum relative variance around $10^7$, whereas the slow-fast version remains around a much controlled range of around $10^1$.

# G  More qualitative visualizations

In Figures 8, 9, 10, 11, 12 and 13, we visualize the predictions of our proposed method on scenes from ScanNet [13] and Messy Rooms. Left-most columns show instance labels obtained after

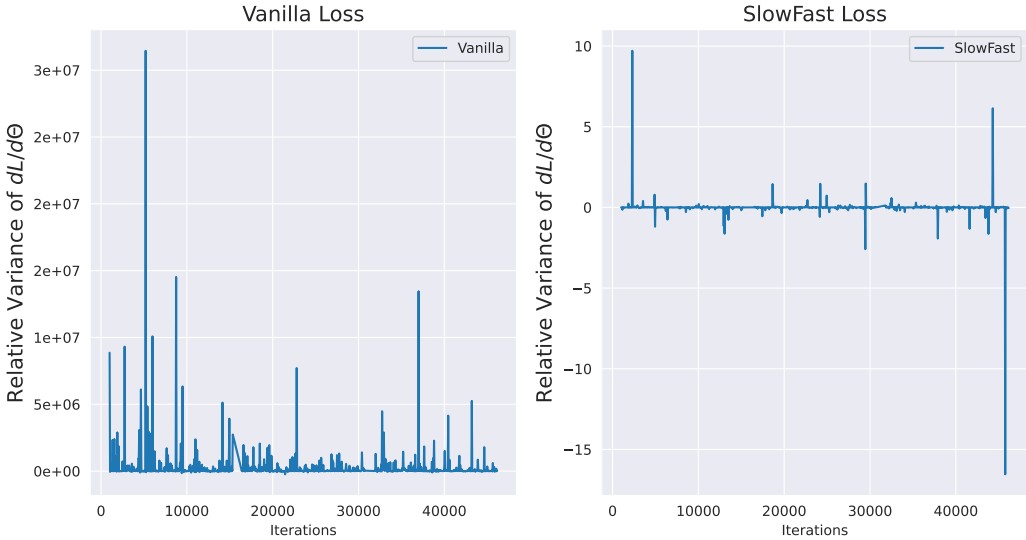

Figure 7: Relative variance (i.e. $Var(\cdot)/Mean(\cdot)$) in loss gradients w.r.t. embeddings (*i.e.* $dL/d\Theta$).

clustering, which as we can see are consistent across different views. To understand how well the embeddings are clustered, we visualize heatmaps of distance of rendered embeddings from cluster-centroids. Specifically, we choose $4$ centroids, and for each centroid $c_i$ and each pixel $u$, we plot $H(u) = -\log(\|\theta_u - c_i\|)$ normalized to $[0, 1]$, where $\theta_u$ is the rendered embedding.

Note that, instance labels are only computed for pixels belonging to the "*thing*" semantic categories (as predicted by the semantic field)[2]. The "*stuff*" pixels are masked out.

As can be seen in all these visualizations, the heatmaps are peaked at the corresponding object locations and close to zero elsewhere which indicates the embeddings are compactly clustered around the corresponding centroid for each object. In Fig. 13, we can see that even in a scene with 50 objects, the embeddings for each instance are distinctly separable.

---

[2]The thing and stuff categories for our Messy Rooms dataset are simply "foreground" and "background".

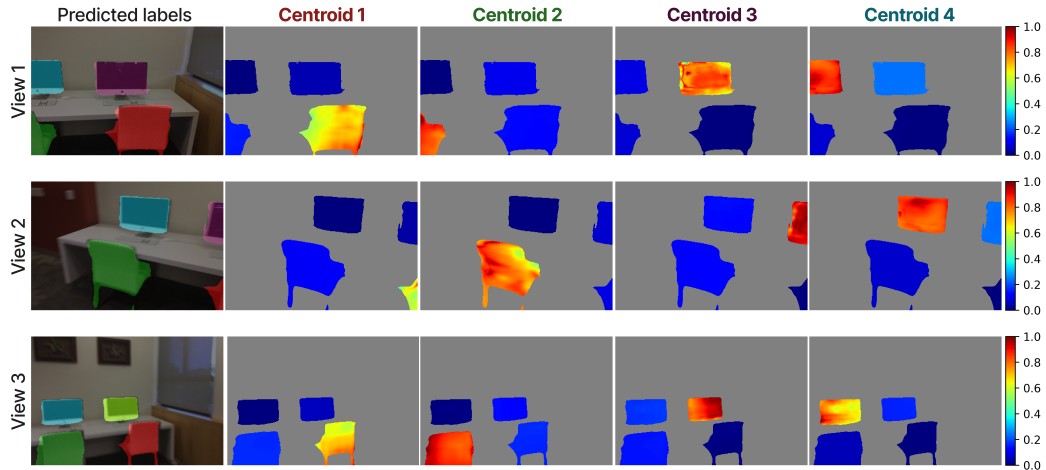

Figure 8: ScanNet scene0300_01: Visualized instance segmentation and clustering heatmaps.

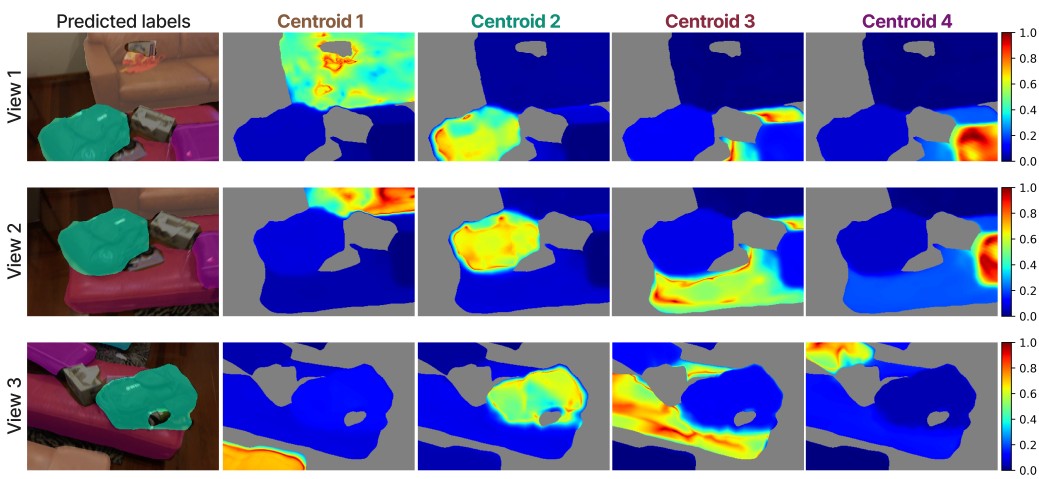

Figure 9: ScanNet scene0050_02: Visualized instance segmentation and clustering heatmaps.

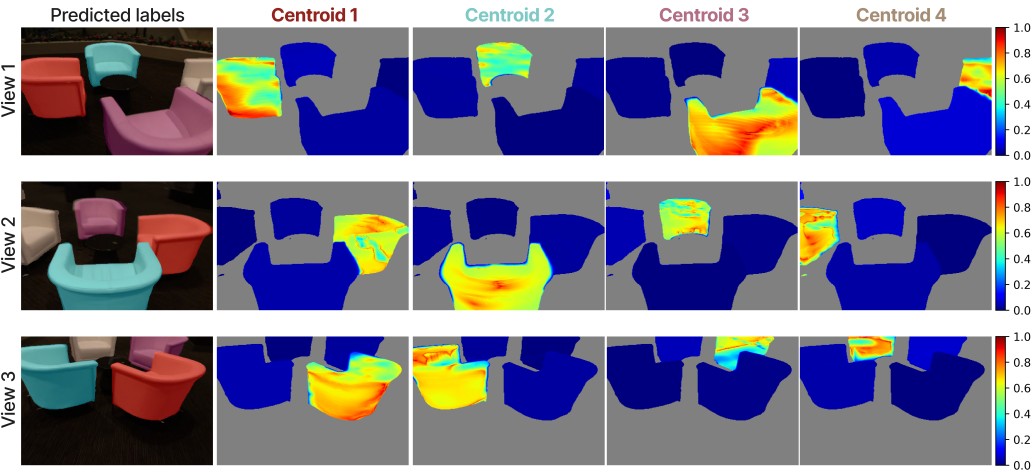

Figure 10: ScanNet scene0423_02: Visualized instance segmentation and clustering heatmaps.

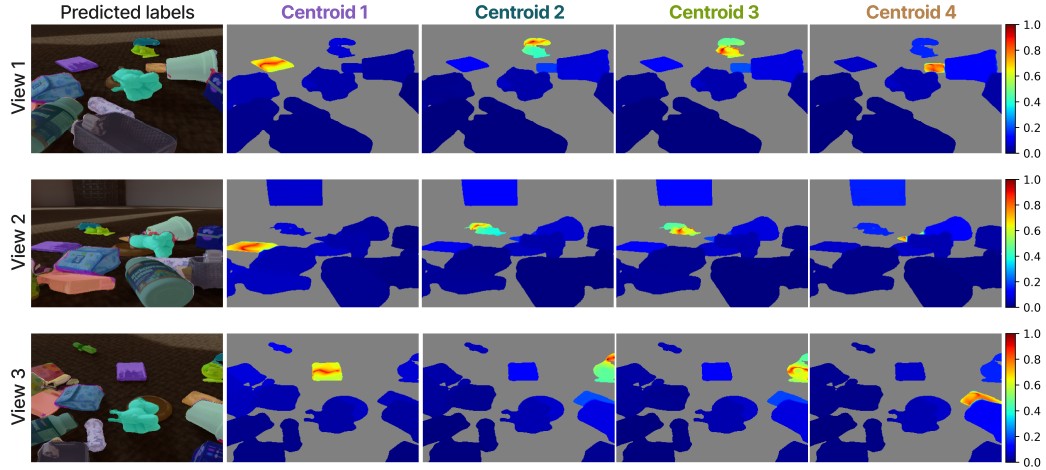

Figure 11: Messy Rooms large_corridor_25: Visualized instance segmentation and clustering heatmaps.

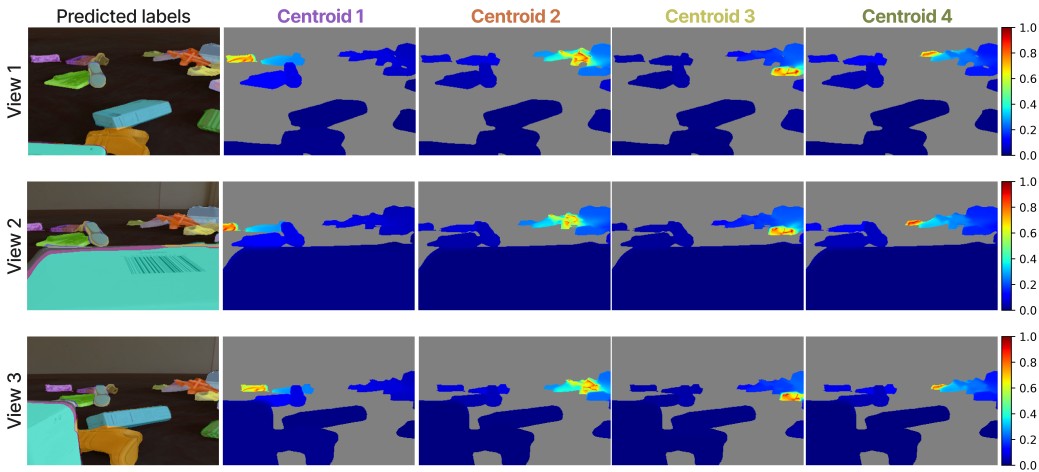

Figure 12: Messy Rooms old_room_25: Visualized instance segmentation and clustering heatmaps.

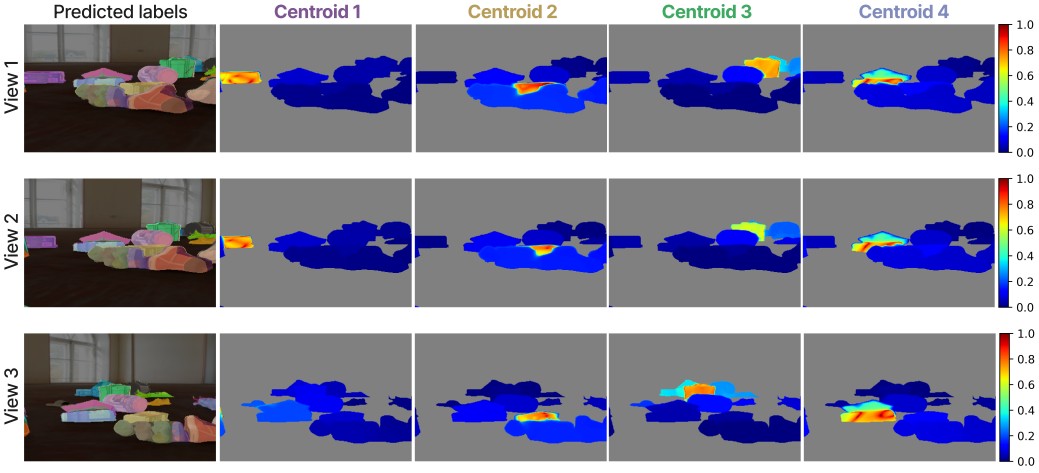

Figure 13: Messy Rooms old_room_50: Visualized instance segmentation and clustering heatmaps.

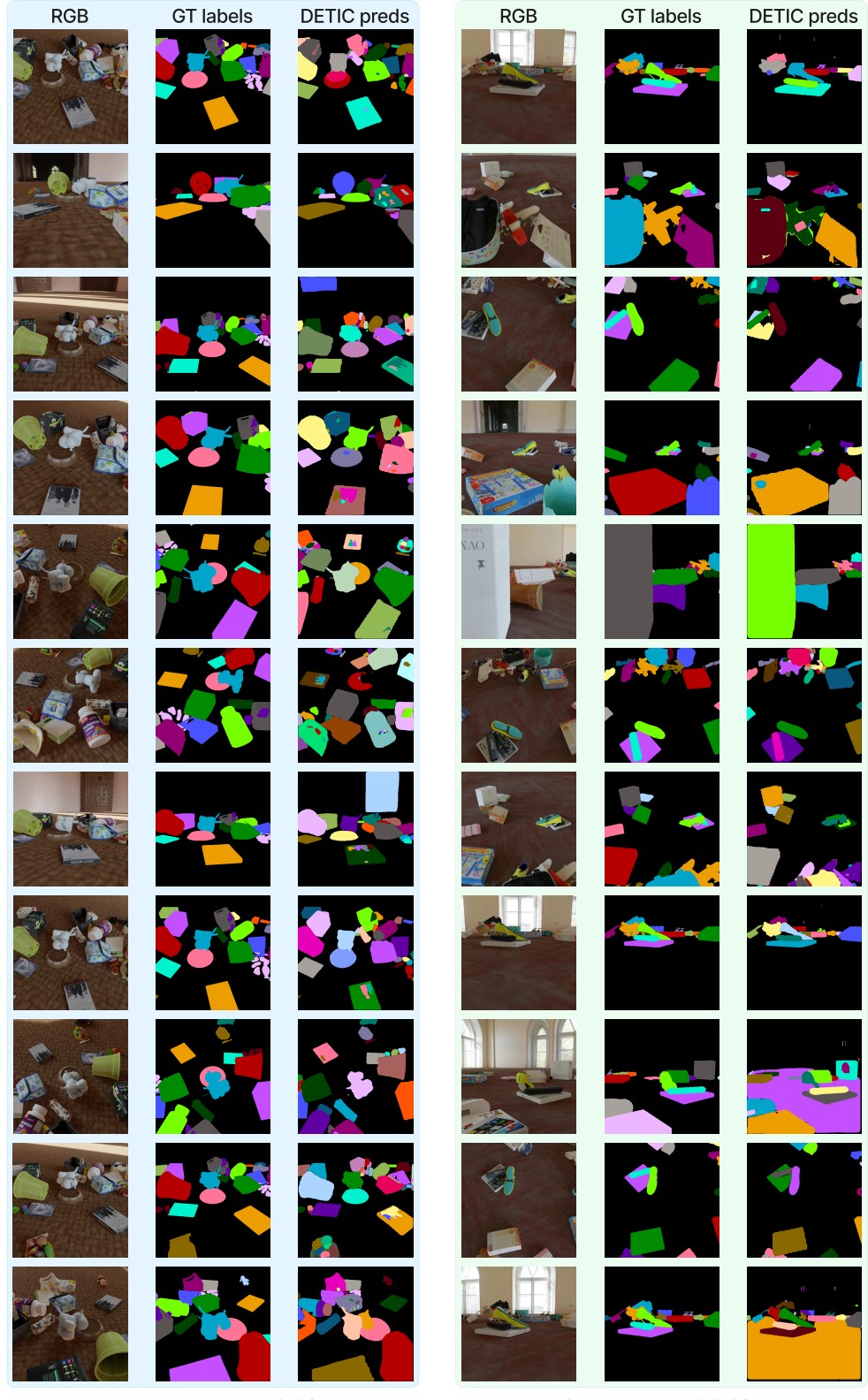

(a) `large_corridor`: 25 objects      (b) `old_room`: 25 objects

Figure 14: Illustrative examples from Messy Rooms dataset. Here, we show scenes with 25 objects.

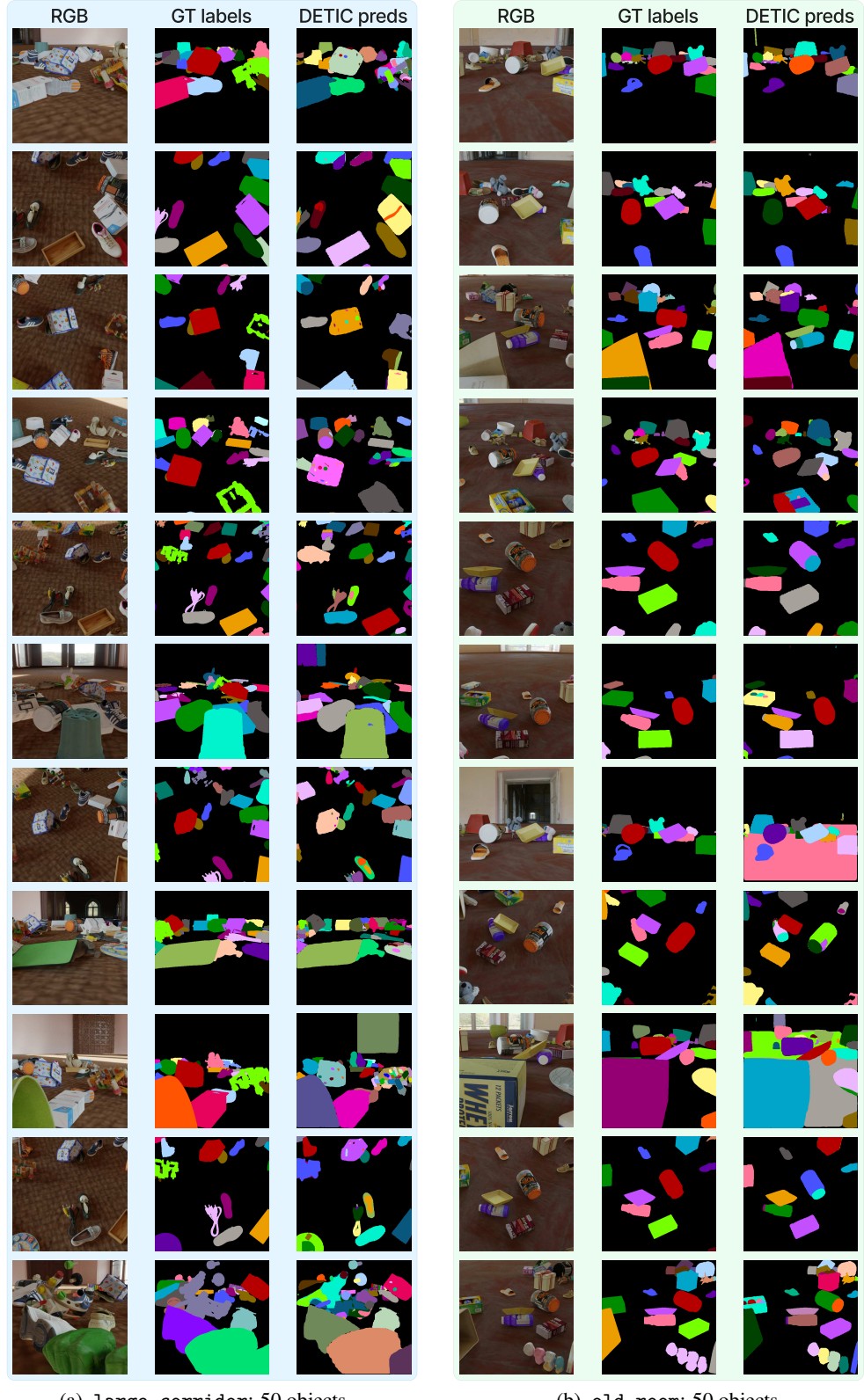

(a) `large_corridor`: 50 objects        (b) `old_room`: 50 objects

Figure 15: Illustrative examples from Messy Rooms dataset. Here, we show scenes with 50 objects.

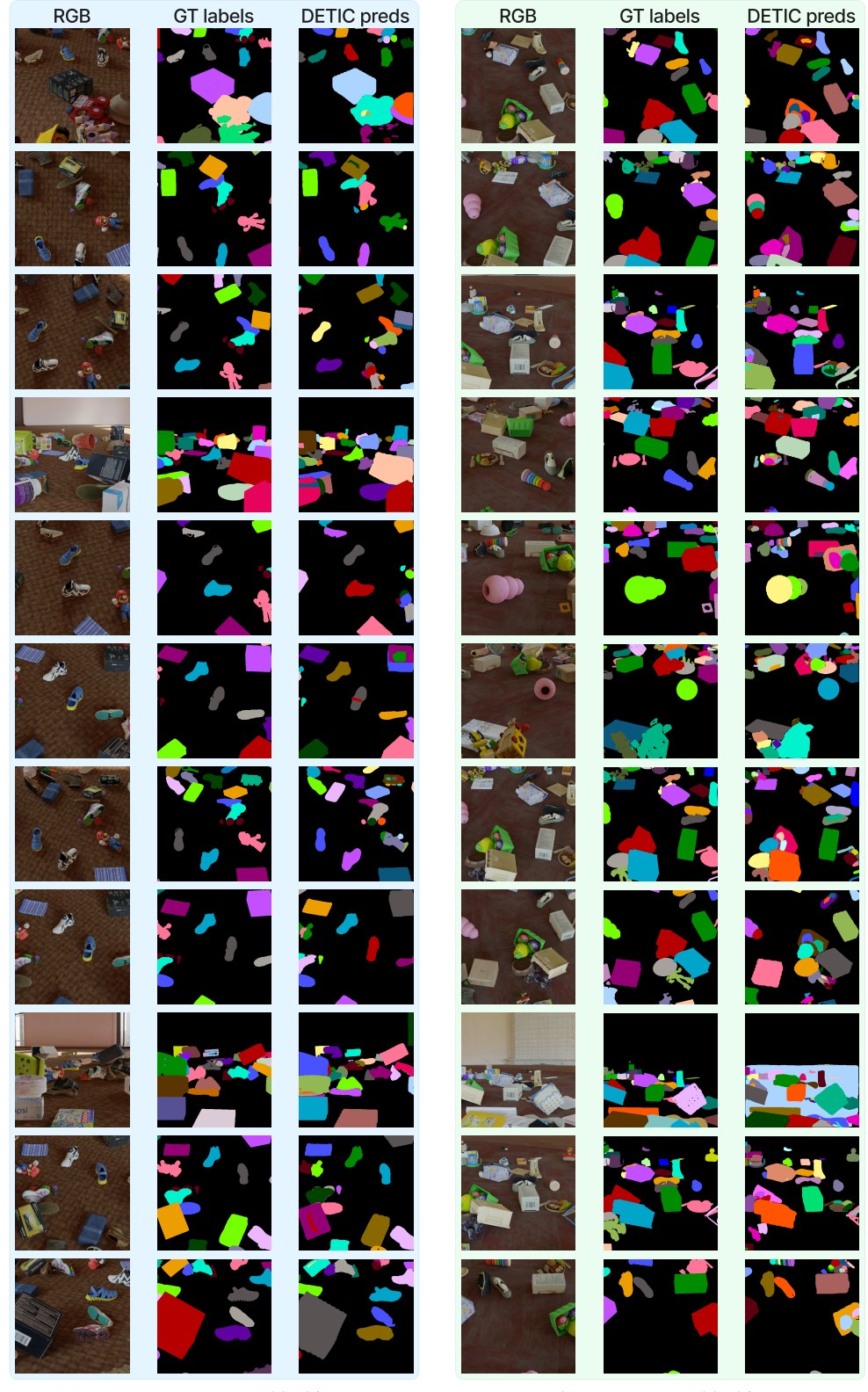

(a) `large_corridor`: 100 objects       (b) `old_room`: 100 objects

Figure 16: Illustrative examples from Messy Rooms dataset. Here, we show scenes with 100 objects.

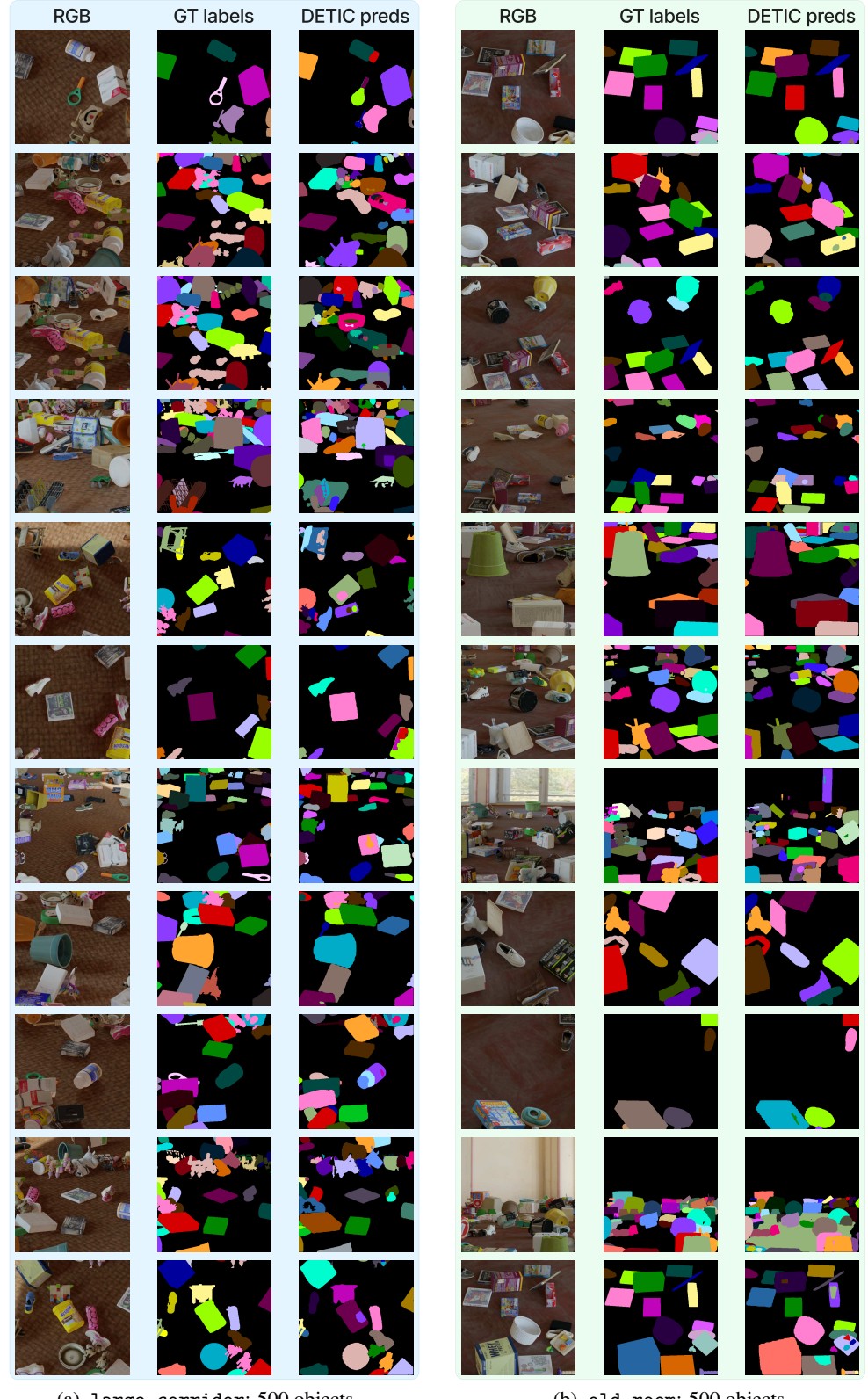

(a) `large_corridor`: 500 objects          (b) `old_room`: 500 objects

Figure 17: Illustrative examples from Messy Rooms dataset. Here, we show scenes with 500 objects.

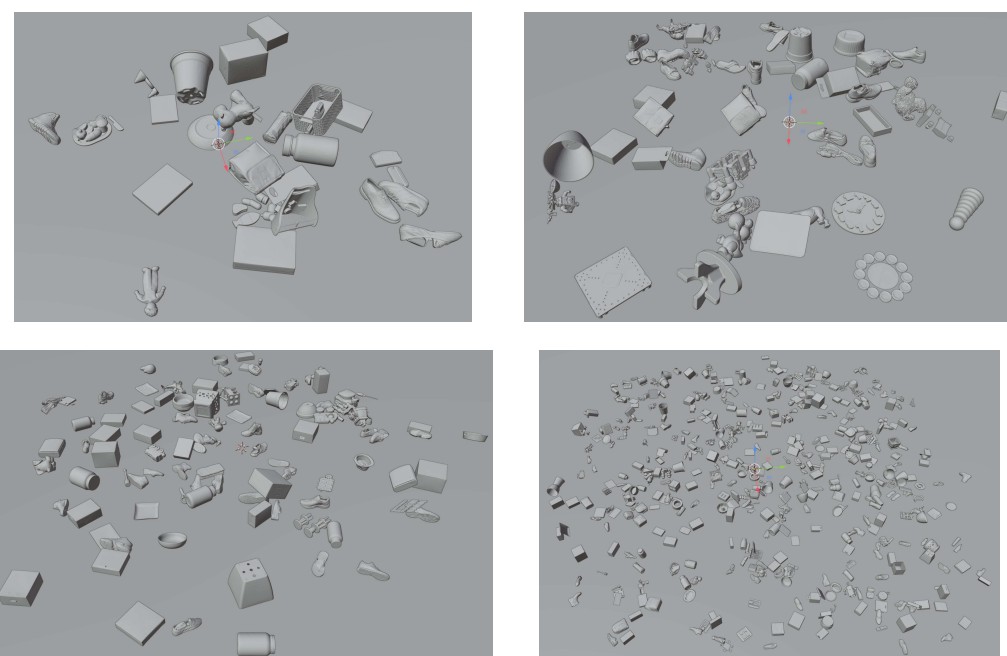

(a) Scenes with `large_corridor` environment. Top left: 25 objects. Top right: 50 objects. Bottom left: 100 objects. Bottom right: 500 objects.

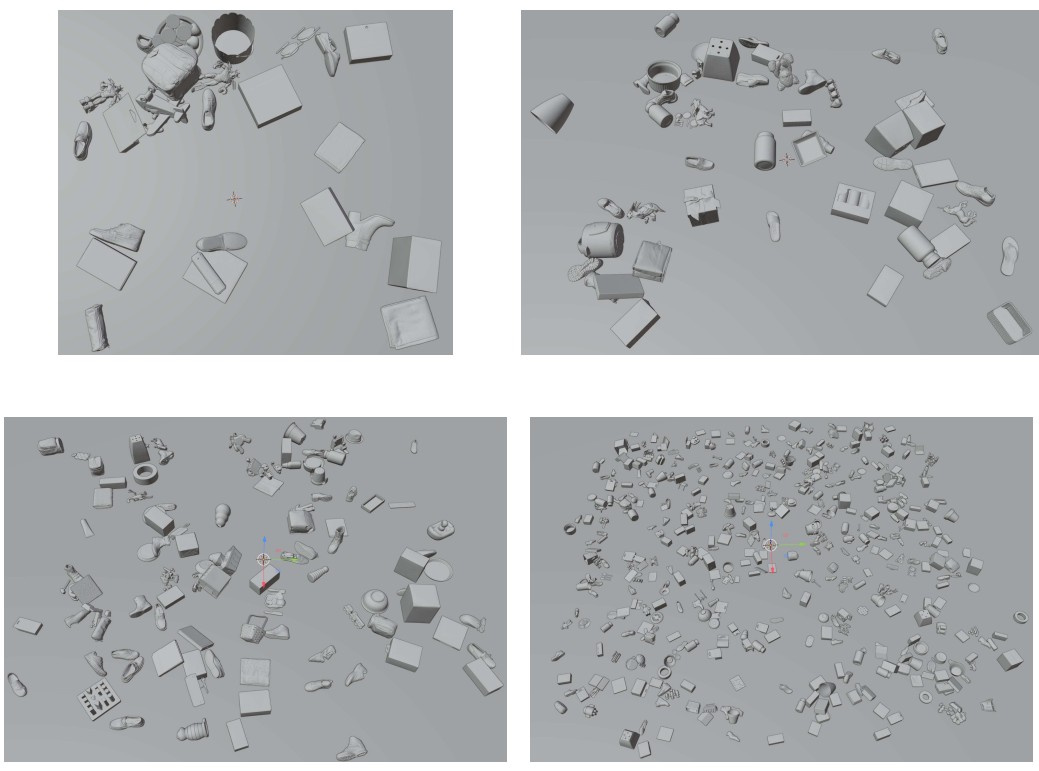

(b) Scenes with `old_room` environment. Top left: 25 objects. Top right: 50 objects. Bottom left: 100 objects. Bottom right: 500 objects.

Figure 18: We show (using Blender [12]) the actual 3D scenes *without texture* that are used to render/generate the Messy Rooms scenes. *Note that the surface area of the scene is increased proportionally to the number of objects.*

