# OpenReview forum: "Contrastive Lift: 3D Object Instance Segmentation by Slow-Fast Contrastive Fusion"
_NeurIPS.cc/2023/Conference — NeurIPS 2023 spotlight_

### Official Review · Reviewer_RRt9 · 2023-06-28

**Soundness:** 2 fair
**Presentation:** 2 fair
**Contribution:** 3 good
**Rating:** 5
**Confidence:** 3

**Summary:**

This paper aims to solve the task of 3D instance segmentation by leveraging pre-trained 2D instance segmentation models. The authors propose a novel approach to lift 2D segments to 3D via a neural field. This idea is not completely new [38, 51], but the authors propose a contrastive loss that replaces the Hungarian-algorithm-based loss used in [38]. Moreover, the authors propose a synthetic dataset where the number of objects can be controlled, and show that the Hungarian-algorithm-based loss slows down substantially as the number of objects increase. Finally, the authors augment the contrastive loss with a momentum-teacher component (similar to [16]).

_[16] Olivier J Hénaff, Skanda Koppula, Evan Shelhamer, Daniel Zoran, Andrew Jaegle, Andrew Zisserman,
João Carreira, and Relja Arandjelović. Object discovery and representation networks. In Computer
Vision–ECCV 2022: 17th European Conference, Tel Aviv, Israel, October 23–27, 2022, Proceedings, Part
XXVII, pages 123–143. Springer, 2022._

_[38] Yawar Siddiqui, Lorenzo Porzi, Samuel Rota Bulò, Norman Müller, Matthias Nießner, Angela Dai, and Peter Kontschieder. Panoptic lifting for 3d scene understanding with neural fields. arXiv.cs, abs/2212.09802,
2022._

_[51] Suhani Vora, Noha Radwan, Klaus Greff, Henning Meyer, Kyle Genova, Mehdi S. M. Sajjadi, Etienne Pot,
Andrea Tagliasacchi, and Daniel Duckworth. NeSF: Neural semantic fields for generalizable semantic
segmentation of 3D scenes. arXiv.cs, abs/2111.13260, 2021_

**Strengths:**

S1) The proposed scheme to deal with instance label ambiguity is novel.

S2) The paper is mostly well-written.

S3) The authors show that the proposed contrastive momentum-teacher loss gives good performance and is computationally lighter than the Hungarian-algorithm-based loss.

S4) The problem of adapting 2D instance segmentation methods to 3D is useful to real-world applications.

**Weaknesses:**

W1) The notation has some issues, is confusing in some places, and deviates a bit from prior work.
- It is not common to let $y$ represent a mapping. It is common to say that $f$ is a mapping such that $y = f(x)$. Same goes for $Y$. The label $y(u)$ is sometimes referred to as a label and sometimes as a function, which is a bit confusing.
- I think $u$ should be the location of a pixel, rather than the actual pixel (which would be in $\mathbb{R}^3$).
- $Y(x)$ is not presented as a function of $x$, but as a function of $u$.
- $\Theta$ is introduced as both a mapping $\Theta: \mathbb{R}^2\rightarrow \mathbb{R}^D$ and $\Theta: \mathbb{R}^3\rightarrow \mathbb{R}^D$.
- Also $\rho$, $c$, and $\Theta$ are introduced as both mappings and actual elements of the codomain of the mapping.
- Usually, $x$ and $d$ are used for 3D location and ray direction. While it is not a crime to change notation, I cannot see a reason for replacing $d$ with $v$.
- Equation (1) does not link to the input to the nerf ($x$ and $v$). The link can only be found in the text in l128-l129. It would be easier to understand if $R$ was actually described as a function of $u$.

W2) The name Slow-Fast is confusing.
- The loss component name slow-fast could have been named _student-teacher_ or something with _exponential moving average_. Currently, the name makes it easy to confuse with Slow-Fast [1001].

W3) The prior work [38] also adapts 2D instance segmentation models to 3D instance segmentation using NeRFs. The proposed approach replaces the matching-based loss and the "slow-fast"-component seems necessary for good performance. Is the comparison to [38] completely fair? Except for the proposed changes, are all other things equal, e.g., the underlying instance 2D segmentation approach, NeRF architecture, or training scheme? I find this difficult to tell from the paper text. It is clear that the proposed approach yields good performance and trains faster than [38], but it is not clear whether this is purely do to the proposed changes.

Minor remarks
- "much smaller" on l62 should probably be removed.


_[1001] Feichtenhofer, Christoph, et al. "Slowfast networks for video recognition." Proceedings of the IEEE/CVF international conference on computer vision. 2019._

**Questions:**

Q1) A prior work [1002] might be relevant to the 3D semantic segmentation background. The references tackles that problem and predates the references [23, 29, 51, 57].

Q2) How does the proposed approach compare to [38]? See W3.

Q3) What does the statement on l290 mean? How does the proposed approach improve noisy 2D semgmentations?

Q4) On l291, what does it mean for 3D reconstruction to work properly?

_[1002] Lawin, Felix Järemo, et al. "Deep projective 3D semantic segmentation." Computer Analysis of Images and Patterns: 17th International Conference, CAIP 2017, Ystad, Sweden, August 22-24, 2017, Proceedings, Part I 17. Springer International Publishing, 2017._

**Limitations:**

The authors provide a discussion that clarifies some important aspects, for instance that the proposed approach supports only static scenes.

---

> ### Author Rebuttal · Authors · 2023-08-07
>
> Thank you for your thoughtful comments which have helped us improve our paper!
>
> ---
>
> **Response to Weakness 1**: We are thankful for the reviewer's detailed feedback on the notation and presentation.
> * We will clarify that $u$ is the pixel location.
> * $\Theta$ should be $\Theta:\mathbb{R}^3\rightarrow\mathbb{R}^D$. And $\theta$, which is defined as $\mathcal{R}(u|\Theta,\rho,\pi)$ in Equation (3), should be $\theta:\mathbb{R}^2\rightarrow\mathbb{R}^D$. We will correct this typo.
> * We acknowledge the concern regarding the dual usage of {$\rho, c, \Theta$} as both mappings and codomain elements. Our intention was to maintain notation for ease of understanding. We'll ensure to clarify the context whenever dual usage occurs, aiming to minimize any confusion.
> * Thank you for the suggestion. We will change the notation to use $d$ for viewing direction, as in common in the literature.
> * We will ensure that the relationships between equations, especially Equation (1), and their corresponding inputs are clearly stated and explained.
>
> ---
>
> **Response to Weakness 2**: We acknowledge that the term "slow-fast" is a bit overloaded (e.g. *Feichtenhofer, Christoph, et al. "Slowfast networks for video recognition."*). However, as described in L149-154, we use a *slowly-updated* field to compute the centroid embeddings and encourage compact clustering in the *fast* field, which is a key element of our approach. We hope that this explanation in the paper can justify the usage of the term "slow-fast contrastive fusion" in our method.
>
> ---
>
> **Response to Weakness 3**: We have made every effort to ensure that the comparison to Panoptic Lifting [38] is completely fair. This can be noted as follows:
> 1. We purposely use the same underlying architecture as [38] so that we can fairly compare Panoptic Lifting’s matching-based instance loss with our proposed “slow-fast” formulation (*while keeping all other things equal*).
> 2. Apart from the instance segmentation related losses (namely, matching-based loss for [38] and our slow-fast loss), all the other losses and training scheme are identical to [38].
> 3. We use the same underlying 2D panoptic (semantic+instance) segmenter for all the methods (which includes Panoptic Lifting [38] and our method). For example, we use Mask2Former [7] for experiments with ScanNet, Replica and Hypersim datasets. And use Detic [59] for experiments with our Messy Rooms dataset.
>
> We will highlight these aspects in the revised paper so that they are clear to the reader.
>
> ---
>
> **Response to minor remark**: Thank you. We will remove the phrase “much smaller” and only state that D<<L, which is more clear.
>
> ---
>
> **Response to Question 1**: Thank you for bringing this method to our attention. It is definitely an influential work in the direction 2D-to-3D semantic fusion to achieve 3D segmentation. We will include this in the discussion of related works.
>
> **Response to Question 2**: Please refer to our "**response to weakness 3**" above which clarifies how we ensure a fair comparison with Panoptic Lifting [38].
>
> ---
>
> **Response to Question 3**: The proposed method fuses 2D segments in 3D space, promoting multi-view consistency across frames. By fusing information from multiple views, the approach is able to overcome the limitations of individual noisy 2D segmentations and produce more accurate and robust instance segmentation *in the 3D space*. As a result, when rendering 2D views from the 3D neural field model, we observe an improvement in quality compared to the original 2D segmentations that were used to train the model in the first place. To clarify this further, we quantify this improvement in the Table below. We report the PQ metric achieved on ScanNet [8] by three different 2D segmentation models: MaskFormer [Cheng et al., “Per-Pixel Classification is Not All You Need for Semantic Segmentation”], Detic [59] and Mask2Former [7]. We compare this to the PQ score achieved by our method when trained with each of these 2D models' predictions. We can see that our method improves the Panoptic Quality (PQ) significantly in each case.
>
> | Method | PQ |
> | -------- | -------- |
> | MaskFormer     |  41.1    |
> | Contrastive Lift (trained with *MaskFormer* labels)     |  61.7    |
> | Detic [59]     |   42.0   |
> | Contrastive Lift (trained with *Detic* labels)     |   61.6   |
> | Mask2Former [7]     |  43.6    |
> | Contrastive Lift (trained with *Mask2Former* labels)     |  **62.1** |
>
> ---
>
> **Response to Question 4**: For 3D reconstruction to work properly, it means that the _learned density field_ accurately represents the geometry of the scene. The proposed method fuses the 2D segmentations from multiple views into a 3D neural field, and this fused field is obtained via differentiable volumetric rendering using the density field (denoted as $\rho$ in Section 3). Hence, to ensure multi-view consistency during the fusion, accurate geometry or density field reconstruction is crucial.

---

### Official Review · Reviewer_Vjqu · 2023-07-03

**Soundness:** 3 good
**Presentation:** 3 good
**Contribution:** 3 good
**Rating:** 7
**Confidence:** 4

**Summary:**

This paper introduces a novel Contrastive Lift method for 2D segment lifting to 3D reconstruction and instance segmentation. The authors fuse multiview representations obtained from pre-trained 2D models into a unified 3D neural field. They propose a scalable slow-fast clustering objective function that enables segmenting without an upper bound on the number of objects. Additionally, a new semi-realistic dataset is created to evaluate the proposed method, which demonstrates superior performance compared to previous state-of-the-art approaches on both public datasets and the newly introduced dataset.

**Strengths:**

1. The research motivation and key challenges are generally well illustrated and summarized.
2. The newly constructed framework and dataset fits well within the 3D vision field, particularly the flexible and scalable design for lifting 2D segments to 3D.
3. The proposed method is technically sound and achieves SOTA performance on the newly proposed dataset and challenging scenes on public datasets.

**Weaknesses:**

1. The paper lacks a discussion on the complexity of the proposed method and its potential impact.
2. Ablations on different 2D segmenters are not included in the paper.
3. The analysis of the ablations and their corresponding results is insufficient.

**Questions:**

Please refere to Weaknesses.

**Limitations:**

Please refere to Weaknesses.

---

> ### Author Rebuttal · Authors · 2023-08-07
>
> Thank you for your valuable feedback!
>
>  ---
>
> **Response to Weakness 1**: Lines 278-288 of the main paper provide an empirical analysis of the time complexity and training speed of our method and compares it to the linear-assignment matching-based method. In summary, the complexity of our proposed method is agnostic to the number of objects in the scene, while the compared matching-based method has a $O(K^3)$ complexity where $K$ is a hyperparameter that specifies the maximum number of objects.
>
> ---
>
> **Response to Weakness 2**: Thank you for the suggestion. Based on the reviewer's suggestion, we have conducted experiments with two more 2D instance segmentation models (*MaskFormer [Cheng et al., "Per-Pixel Classification is Not All You Need for Semantic Segmentation"] and Detic* [59]) in addition to *Mask2Former* [7].
>
> The Table below demonstrates the Panoptic Quality (PQ) metric achieved on the ScanNet dataset [8] by each of these 2D models and by our method when trained with predictions from these 2D segmentation models. In all cases, we can see that our method improves the PQ metric significantly.
>
> | Method | PQ |
> | -------- | -------- |
> | MaskFormer     |  41.1    |
> | Contrastive Lift (trained with *MaskFormer* labels)     |  61.7    |
> | Detic [59]     |   42.0   |
> | Contrastive Lift (trained with *Detic* labels)     |   61.6   |
> | Mask2Former [7]     |  43.6    |
> | Contrastive Lift (trained with *Mask2Former* labels)     |  **62.1** |
>
> ---
>
> **Response to Weakness 3**: Thank you for the suggestion. We will include the above ablation regarding the effect of different 2D segmenter models in the revised paper. We shall also expand the discussion in the ablation studies included in Section 5.2 of the main paper as well as Sections 3 and 5 of the supplementary material.

---

> > ### Comment · Reviewer_Vjqu · 2023-08-21
> > **Post-rebuttal comment**
> >
> > Thanks for the rebuttal. My concerns have been addressed in a satisfying way. Thus I am happy to lift the rating to Accept.

---

### Official Review · Reviewer_4dod · 2023-07-05

**Soundness:** 3 good
**Presentation:** 3 good
**Contribution:** 3 good
**Rating:** 7
**Confidence:** 3

**Summary:**

This paper utilizes contrastive learning for lifting 2D segments to 3D and fuses the learned embedding by means of a neural field representation, namely Contrastive Lift. The authors further propose a slow-fast clustering objective function, which makes the method scalable for scenes with a large number of objects. To further validate the ability of the method, this paper also introduces a new dataset, Messi Rooms, which includes up to 500 objects as a benchmark for instance segmentation with a large number of objects. The experiments show that the proposed approach outperforms former SOTA on ScanNet, Hypersim, Replica, and Messy Rooms.

**Strengths:**

- The proposed approach employs a low-dimensional Euclidean embedding to represent a 3D instance. The dimensionality D is far less than the number of objects L, making the approach more efficient and easily extended to larger numbers of objects. Most importantly, using the 3D instance embedding implicitly ensures multi-view consistency. It avoids the assignment problem that exists in Panoptic Lifting [43].

- Using contrastive learning and the clustering strategy makes the proposed approach independent of the number of objects, which is more suitable for scenes with a large number of objects.

- The slow-fast contrastive learning is scalable for different object numbers and stabilizes the training phase. And the proposed concentration loss ensures the concentration of embeddings within the same cluster. It improves the more complete instance segmentation results.

- The proposed semi-realistic dataset, Messy Rooms, provides a novel benchmark for testing the performance on scenes with large object numbers.


**Weaknesses:**

- In Tab. 1 and Tab.2, the metric used for evaluation only uses PQ^scene^, which is cherry-picked. In Panoptic Lifting [43], mIoU and PSNR are also used for evaluation. A comprehensive comparison according to different metrics should be included.

- For semantic segmentation, Contrastive Lift should append a new branch to predict the semantic labels and be supervised by the segment consistency loss in [43]. I suppose that the model should be trained specifically for semantic segmentation. Panoptic Lifting can predict semantic and instance labels simultaneously. I am curious about whether the additional supervision on semantic labels would influence the instance segmentation results. Please explain it.

**Questions:**

- The random partition of pixels into two non-overlapping sets needs further explanation. (Randomly choose pixels? Randomly divide images with pre-defined boundaries? What is the proportion between two sets? etc.)

**Limitations:**

The authors have addressed some limitations in Sec. 6. There is no negative social impact in this work.

---

> ### Author Rebuttal · Authors · 2023-08-07
>
> Thanks for providing feedback and taking the time to review our work!
>
> ---
>
> **Response to Weakness 1**: We report the mIoU and PSNR of our method (and also for Panoptic Lifting and SemanticNeRF) in Table 2 of the **supplementary material**. In Tables 1 and 2 of the main paper, we only report PQ$^\text{scene}$ for brevity, as it is the most relevant metric for assessing instance segmentation quality, which is the main contribution of our approach.
>
> Since the semantic field and density/color field architecture of our method is identical to Panoptic Lifting [38], we expect to achieve similar mIoU (semantic segmentation performance) and PSNR (view synthesis performance) as [38]. This can be verified from Table 2 of supplementary material.
>
> ---
>
> **Response to Weakness 2**: We would like to clarify the following points about our architecture and loss functions:
> 1. As described in lines 168-176, our architecture has separate branches for density, color, semantic and instance prediction. Our architecture is the same as the one used in Panoptic Lifting, which also has these 4 separate branches. We purposely use the same underlying architecture so that we can fairly compare Panoptic Lifting’s matching-based instance loss with our proposed “slow-fast” loss formulation with instance embeddings (*while keeping all other things equal*).
> 2. We do have supervision on semantic labels, including the segment consistency loss (line 166 in the main paper). We will make this part more clear in the revised paper.
> 3. Note that the gradients from the semantic loss(es) do NOT propagate to the instance branch (and vice versa), as these are parallel/independent branches. That being said, supervising semantic labels does not affect instance segmentation performance but it affects panoptic segmentation which considers both semantic and instance predictions.
>
> ---
>
> **Response to Questions**: These pixels are chosen purely randomly from $\Omega$. Both of these non-overlapping sets are of equal size. To be more precise: say we want to sample two non-overlapping sets, each of size $N$. We first sample a batch of $2N$ pixels from $\Omega$ and then simply split it into two sets of size $N$.

---

> > ### Comment · Reviewer_4dod · 2023-08-16
> > **Thanks for the rebuttal**
> >
> > The authors address my concerns in the rebuttal. After reading the authors' replies and other reviews, I will keep my rating.

---

### Official Review · Reviewer_KUUX · 2023-07-06

**Soundness:** 3 good
**Presentation:** 4 excellent
**Contribution:** 3 good
**Rating:** 7
**Confidence:** 4

**Summary:**

The proposed method tries to solve the problem of reconstructing a 3D scene together with the underlying instance segmentation. Prior work required either GT tracking data or concurrent work a less efficient way to assign instances. From a set of images a Neural Radiance Field is reconstructed together with a feature field, that represents an embedding of the instance. Instance embeddings are guided by a contrastive loss function, that pushes embeddings that are projected into pixels from the same segment in a 2D segmentation mask closer and embeddings projecting into different masks apart. To improve the stability of the training, the authors propose an additional loss with a jointly trained slowly-updated instance embedding field updated with a moving average over the parameters of the faster field instead of SGD.
Instances are later computed by clustering embeddings, which is supported by the third loss term, which uses an average embedding from the slowly updated field to penalize the difference for the fast-field predictions. Specific values in the embedding vector are assigned to semantic classes for semantic segmentation and are directly supervised with the 2D semantic maps.
Additionally, the authors propose a novel dataset with up to 500 objects for evaluating future 3D instance segmentation

**Strengths:**

So far 3D instance segmentation methods, such as Neural Panoptic Fields (referenced by the authors as well) required a tracking algorithm or GT tracking data to reconstruct instance labels of the 3D scene and this method presents a light, optimization-based approach, that directly learns an instance embedding from semantic maps through alignment.

The authors describe their method in a way that is understandable and design choices, especially for the loss function and the learning paradigm are reasonable and justified by ablation and additional experiments.

In general, this is a well-designed method that leverages the current state of the art and adds interesting new components to allow a jointly learning of the radiance field, instance and semantic embedding.

A big plus of the presented method is the additional dataset with up to 500 objects and the submission of the code, that allows the reproducibility of the results.


**Weaknesses:**

While the presented evaluation of the proposed dataset shows a clear advantage over state-of-the-art methods on existing and their own synthetic dataset, as well as ScanNet, methods like Panoptic Fields have also shown results on complex outdoor scenes, such as the KITTI dataset. Therefore an evaluation in such a complex outdoor setting would further strengthen the paper and usability in future work.

**Questions:**

- It seems like a fixed number of clusters is given or how is the number of clusters decided?

**Limitations:**

The authors address most of the limitations I can think of and ablate the use of different methods they rely on, such as the clustering method in the supplement.

---

> ### Author Rebuttal · Authors · 2023-08-07
>
> Thank you for taking the time to study our work and provide thoughtful feedback!
>
> **Response to Weaknesses:** Thank you for suggesting the idea to assess the performance of our approach on outdoor scenes, e.g. *KITTI* or *KITTI-360* datasets. We agree that it would strengthen the paper. Unfortunately, we could not produce the results on the KITTI dataset in the given limited time for rebuttal, but we will include results on these outdoor scenes in the final version of the paper. As you have already pointed out, we show in Table 1 that our method outperforms Panoptic Neural Fields on indoor scenes.
>
> **Response to Questions:** We ***do not*** assume knowledge of the number of clusters. Instead, we use HDBSCAN (Hierarchical DBSCAN) [31] which is a density-based clustering algorithm. HDBSCAN does not require the number of clusters to be known. It computes a “mutual reachability distance” matrix between all pairs of datapoints and then uses a single linkage algorithm to find clusters in the data. The only hyperparameter in HDBSCAN is the minimum cluster size, which can be set to any reasonable value (e.g. $10^3$ if we have $10^5$ datapoints) or can be grid-searched on a fraction of the training dataset.

---

> > ### Comment · Reviewer_KUUX · 2023-08-21
> >
> > Thanks for the additional comments and clarification! Given there is not much inconsistencies in the reviews, I will keep my rating and recommend acceptance.

---

### Official Review · Reviewer_zMUp · 2023-07-06

**Soundness:** 3 good
**Presentation:** 3 good
**Contribution:** 3 good
**Rating:** 7
**Confidence:** 4

**Summary:**

The paper studies the 3D object instance segmentation inside a 3D NeRF space. Specifically, to train the model, conservative loss between features generated by slow and fast NeRF models are computed to 1) maximize the feature distance between different semantic regions, 2) minimize the feature distance within the same semantic regions. Also, a dataset is introduced namely Messy Room, which consists of rendering real captured objects from Google Scanned Objects. Experiments on the dataset show a reasonable improvement in comparison to baselines.

**Strengths:**

The proposed contrastive learning for 3D semantic segmentation on NeRF is elegant and novel. I would believe such a structure is considered to be better in comparison to previous works that directly output the semantic labels from the NeRF network. Also, the introduced messy rooms dataset is believed to be useful for the community, despite the dataset is partially synthetic. The overall performance improvements from the baseline of the proposed method are not huge but still reasonable.

**Weaknesses:**

1) The proposed method is only evaluated on the half-synthetic dataset with small objects on the table. It is necessary to evaluate the proposed method on some real images, at least qualitatively.

2) The performance of the segmentations before lifting is not reported in the experiment section, it is unclear how the proposed method improves from the 2D segmentation.

3) The contrastive training pipeline is partially similar to this work [1], it would be better to include it in discussion.

[1] Bai, Y., Wang, A., Kortylewski, A., & Yuille, A. (2020). Coke: Localized contrastive learning for robust keypoint detection. arXiv preprint arXiv:2009.14115.

**Questions:**

1) How the ground plane is determined and set in the messy Room dataset?

2) The author mentions in line 148 that "gradients with high variance". What is a high variance on gradients? The author should quantitatively study how the variance is controlled using the proposed SF loss in comparison to vanilla contrastive loss.

3) A clustering operator is conducted on the feature vectors for segmentation, is this similar to this work? [2]

4) In the visualization, how does the trained feature vectors convert into semantic labels?

5) Some typos: in figure 2 L_cntr but in equation 5 L_conc.


[2] Yu, Q., Wang, H., Qiao, S., Collins, M., Zhu, Y., Adam, H., ... & Chen, L. C. (2022, October). k-means Mask Transformer. In European Conference on Computer Vision (pp. 288-307). Cham: Springer Nature Switzerland.


**Limitations:**

The authors include and discuss limitations of this work. There seems no potential negative societal impact.

---

> ### Author Rebuttal · Authors · 2023-08-07
>
> Thank you for taking the time to read our paper and provide feedback!
>
> ---
>
> **Response to Weakness 1**: We do evaluate on real scenes from ScanNet [8] dataset (please refer L211-212 and Table 1). We also evaluate using Replica [41], which comprises real 3D scans, and Hypersim [36], which is synthetic but designed to match the complexity of real scenes. In addition to quantitative results in Tables 1,2 and 3, Figures 1,2,3 in the supplementary show visualizations on ScanNet scenes. We will add more visualizations from these datasets in the final version.
>
> ---
>
> **Response to Weakness 2**: The performance before lifting is not shown because these instance segmentations (obtained from a 2D segmenter) are *not consistent (aka tracked) across frames/views*. Therefore, the $PQ^{scene}$ metric for these 2D methods is very low, because the metric is designed to reflect the cross-frame consistency of the predictions.
>
> **Following the reviewer’s suggestion, we evaluate performance of 2D segmentations (i.e., before lifting) in 2 ways:**
> 1. Table 1 (see below) reports the _frame-level_ PQ (Panoptic Quality) score for both the 2D segmenter and our method. Note that PQ only evaluates segmentation quality frame-by-frame, regardless of inter-frame consistency. The table shows that our method significantly improves the frame-level 2D segmentation quality.
> 2. Table 2 (see below) reports the PQ$^{\text{scene}}$ metric, which considers the consistency of instance segmentations across frames. To achieve consistency, we post-process the 2D segmenter's predictions using Hungarian Matching for cross-frame tracking as follows:
>     * **w/ Hungarian matching (2D IoU):** Given sets of predicted segments ($P_i$ and $P_{i+1}$) from consecutive frames, compute IoU matrix by comparing all segment pairs in $P_i\times P_{i+1}$. Apply Hungarian matching to the IoU matrix to associate instance segments across frames.
>     * **w/ Hungarian matching based on IoU after depth-aware pose-warping**: Use *ground-truth* pose and depth for warping $(i+1)$-th frame's segmentation to frame $i$. Compute IoU matrix using warped segmentations and apply Hungarian matching.
>     * **w/ Hungarian matching using the 3D ground-truth pointcloud**: Using only consecutive frames leads to errors in long-range tracking. To address this, starting from 1st frame, un-project 2D segmentations into a 3D point cloud. Iteratively fuse these segments in 3D using Hungarian matching. This way, segments from *all preceding frames* along with 3D information are used for tracking.
>
> We note that the last two baselines use the _3D ground-truth_ for matching. The table below shows that, even when 3D information is used, our method still significantly outperforms the baseline 2D segmentation approach.
>
> **Table 1**:
> | Method                         | PQ (on ScanNet [8]) |
> | ------------------------------ | ------------------: |
> | Mask2Former [7] (2D segmenter) | 43.6                |
> | Contrastive Lift (**ours** trained w/ Mask2Former labels) | **62.1**|
>
> **Table 2**:
> | Method | PQ$^{\text{scene}}$ (on ScanNet) |
> | -------- | --------: |
> | Mask2Former (*M2F*) (non-tracked 2D segmentations)     | 32.3     |
> | M2F + Hungarian matching (2D IoU)     | 33.7     |
> | M2F + Hungarian matching based on IoU after ***depth-aware pose-warping***     | 34.0     |
> | M2F + Hungarian matching using the ***3D ground-truth pointcloud***     | 41.0     |
> | Contrastive Lift (**ours** trained w/ Mask2Former labels)    | **62.3**     |
>
> ---
>
> **Response to Weakness 3**: Thank you for bringing this method to our attention. CoKe is an interesting method that uses contrastive learning along with moving average updates to learn keypoint prototypes. We will include CoKe in the discussion of related methods.
>
> ---
>
> **Response to Question 1:** The Messy Rooms dataset is generated using the Kubric [12] simulator. We configure the "gravity" vector along the "-z" axis and designate the xy-plane as the ground-plane. For details, see the code in `dataset/kubric_panopli_generator_final.py`.
>
> ---
>
> **Response to Question 2:** In L148, we refer to the variance in loss gradients w.r.t. the embedding field ($\Theta$), i.e. variance of $\nabla_{\Theta} L$. Empirically, we compute the relative variance ($\frac{Var(\cdot)}{Mean(\cdot)}$), finding significantly higher values (across training iterations) with the vanilla loss as compared to the slow-fast version. Please see Figure 1 in the attached PDF. The vanilla loss shows spikes with a peak relative variance near $10^7$, while the slow-fast variant maintains around $10^1$.
>
> ---
>
> **Response to Question 3:** *kMaX-DeepLab* learns a pixel-cluster assignment by reformulating cross-attention from a clustering perspective. So, it is indeed similar in spirit to our approach, although it's worth noting that the embeddings (and cluster centers) in our method (defined as $\mathbb{R}^3 \rightarrow \mathbb{R}^D$) are learnt using *differentiable volumetric rendering* from 2D labels. We will include a discussion of *kMaX-DeepLab* (along with “CMT-Deeplab: Clustering Mask Transformers for Panoptic Segmentation”, Yu et al. and “Semi-convolutional Operators for Instance Segmentation”, Novotny et al.) in the revised paper.
>
> ---
>
> **Response to Question 4:** For the ***semantic*** labels, since the features were trained with a cross-entropy loss (similar to [38,57]), we obtain the label as the `argmax` of rendered logits.
> For the ***instance*** labels, as described in lines 177-181 of main paper and in section 2.3 of supplementary material:
> 1.  the rendered instance features are clustered with HDBSCAN [31], an unsupervised density-based clustering algorithm, to obtain cluster centroids.
> 2. for pixels in any novel view, the label of the centroid nearest to the rendered embedding is assigned as the instance label.
>
> ---
>
> **Response to Question 5:** Thank you for pointing this out. It should be $L_{conc}$ in figure 2. We will fix this in the revised paper.

---

> > ### Comment · Reviewer_zMUp · 2023-08-14
> > **Reply to Rebuttal**
> >
> > Thanks for the rebuttal. The rebuttal address most of my concern. I will increase my final rating to accept.

---

### Author Rebuttal · Authors · 2023-08-07

We thank all the reviewers for their time and valuable feedback. We are pleased to see positive responses from all reviewers, who acknowledge the novelty [*zMUp*,*RRt9*], design [*KUUX*,*4dod*,*Vjqu*], efficiency [*4dod*,*RRt9*] and performance [*Vjqu*,*RRt9*] of our approach as well as the usefulness of the proposed dataset [*zMUp*, *KUUX*, *4dod*].

We have answered each reviewer's concerns/questions **separately in the individual responses below**. Additionally, we have attached a PDF that contains one figure which addresses a question from *Reviewer zMUp*.

---

### Decision · Program_Chairs · 2023-09-21

**Decision:**

Accept (spotlight)

**Comment:**

This paper proposes an approach for 3D instance segmentation in Nerfs by lifting multiple 2D views of scenes with available instance segmentation labels. It proposed a novel slow-fast contrastive learning approach to solve the instance association problem, which greatly simplifies the the multi-view assignment problem for different instances across views and also makes it easier to scale up to a large number of object instances. Additionally, a new Messy-Rooms dataset is proposed with many object instances to evaluate the proposed approach.

The paper received the scores (4A, 1BA). All reviewers appreciated the work for its novel and elegant method, clear improvements in quality over the existing state of the art and for appreciated the proposed new dataset as being a valuable contribution to the research community. Most reviewers' concerns were also successfully addressed by the authors' responses.

The AC recognizes the paper's significant contribution to 3D neural instance segmentation and recommends acceptance. Congratulations!

The authors are strongly encouraged to incorporate the changes that they've promised in the rebuttal to the final manuscript.